



# Advanced calibration of magnetometers on spin-stabilized spacecraft based on parameter decoupling

Ferdinand Plaschke[1], Hans-Ulrich Auster[2], David Fischer[1], Karl-Heinz Fornaçon[2], Werner Magnes[1], Ingo Richter[2], Dragos Constantinescu[2], and Yasuhito Narita[1]

[1]Space Research Institute, Austrian Academy of Sciences, Graz, Austria.
[2]Institute for Geophysics and Extraterrestrial Physics, Braunschweig University of Technology, Braunschweig, Germany.

**Correspondence:** Ferdinand Plaschke (ferdinand.plaschke@oeaw.ac.at)

**Abstract.** Magnetometers are key instruments onboard spacecraft that probe the plasma environments of planets and other solar system bodies. The linear conversion of raw magnetometer outputs to fully calibrated magnetic field measurements requires the accurate knowledge of 12 calibration parameters: 6 angles, 3 gain factors, and 3 offset values. The in-flight determination of 8 of those 12 parameters is enormously supported if the spacecraft is spin stabilized, as an incorrect choice of those parameters will lead to systematic spin harmonic disturbances in the calibrated data. We show that published equations and algorithms for the determination of the 8 spin-related parameters are far from optimal, as they do not take into account the physical behavior of science-grade magnetometers and the influence of a varying spacecraft attitude on the in-flight calibration process. Here, we address these issues. Based on decades-long developments and experience in calibration activities at the Braunschweig University of Technology, we introduce advanced calibration equations, parameters, and algorithms. With their help, it is possible to decouple different effects on the calibration parameters, originating from the spacecraft or the magnetometer itself. A key point of the algorithms is the bulk determination of parameters and associated uncertainties. Lowest uncertainties are expected under parameter specific conditions. By application to THEMIS-C magnetometer measurements, we show where these conditions are fulfilled along a highly elliptical orbit around Earth.

## 1 Introduction

The investigation of the plasma environment in the heliosphere, around planets, moons, comets, or other solar system bodies, requires accurate in-situ observations of the magnetic field. Magnetometers on board spacecraft can provide these key measurements if accurately calibrated on ground and in flight. The calibration process delivers the parameters needed to convert raw magnetometer measurements into magnetic field observations $\boldsymbol{B} = (B_x, B_y, B_z)^{\mathrm{T}}$ in physically meaningful coordinate systems and units (usually nanotesla: nT). Commonly, a linear calibration equation is applied for this conversion (e.g., Fornaçon et al., 1999; Balogh et al., 2001b; Auster et al., 2008):

$$\boldsymbol{B} = \mathbf{C} \cdot (\boldsymbol{B}_{\mathrm{S}} - \boldsymbol{O}_{\mathrm{S}}) \tag{1}$$

Here $\boldsymbol{B}_{\mathrm{S}} = (B_{\mathrm{S1}}, B_{\mathrm{S2}}, B_{\mathrm{S3}})^{\mathrm{T}}$ is the raw magnetometer output in non-orthogonal sensor coordinates, $\boldsymbol{O}_{\mathrm{S}}$ corrects for non-vanishing magnetometers outputs in zero ambient fields (so-called offsets, which include spacecraft-generated magnetic fields



at the sensor position), and $\mathbf{C}$ is the $3 \times 3$ coupling matrix. This matrix may have the following form (e.g., Kepko et al., 1996):

$$\mathbf{C} = \begin{pmatrix} \sin\theta_1\cos\phi_1 & \sin\theta_1\sin\phi_1 & \cos\theta_1 \\ \sin\theta_2\cos\phi_2 & \sin\theta_2\sin\phi_2 & \cos\theta_2 \\ \sin\theta_3\cos\phi_3 & \sin\theta_3\sin\phi_3 & \cos\theta_3 \end{pmatrix}^{-1} \cdot \begin{pmatrix} G_{S1} & 0 & 0 \\ 0 & G_{S2} & 0 \\ 0 & 0 & G_{S3} \end{pmatrix} \quad (2)$$

The coupling matrix $\mathbf{C}$ depends on three scaling factors ($G_{S1}$, $G_{S2}$, and $G_{S3}$, also called the gains) and six angles ($\theta_1$, $\theta_2$, $\theta_3$, and $\phi_1$, $\phi_2$, $\phi_3$) which define the directions of the three sensor axes in the orthogonal coordinate system to which $\mathbf{B}$ pertains.

Calibrating a magnetometer means finding the three gains, six angles, and three offset components (i.e., in total 12 parameters) so that $\mathbf{B}$ can accurately be obtained from $\mathbf{B}_S$.

Operating the magnetometer on a spinning spacecraft, instead of on a three-axis stabilized spacecraft, enormously supports the determination of 8 of the 12 calibration parameters. These 8 spin-related parameters are: the two spin plane offset components, five of the six sensor direction angles (all but one defining the rotation about the spin axis), and the ratio of the spin

plane gains. The reason is that an incorrect choice in any of those 8 spin-related parameters leads to the appearance of clear, systematic signals at the spin frequency (also called the first harmonic) and/or at twice the spin frequency (second harmonic) in the de-spun magnetic field measurements. Hence, minimization of these signals can be used to determine the 8 calibration parameters, as described in Farrell et al. (1995) and Kepko et al. (1996).

The other 4 (spin-unrelated) parameters are the absolute gains in the spin plane and along the spin axis, the spin axis offset,

and the angle of rotation of the sensor about the spin axis. Gains and angle can be derived in flight by comparison of magnetic field measurements with the International Geomagnetic Reference Field (IGRF) or the Tsyganenko field models, which are fairly accurate close to Earth (e.g., Thébault et al., 2015; Tsyganenko and Sitnov, 2007). For the determination of the spin axis offset in flight, a list of different methods exists. Typically, the offset is obtained from careful analysis of Alfvénic magnetic field fluctuations, present in the pristine solar wind (e.g., Belcher, 1973; Hedgecock, 1975; Leinweber et al., 2008). If strongly

compressional fluctuations are observed instead of Alfvénic fluctuations, then the mirror mode method may be used (Plaschke and Narita, 2016; Plaschke et al., 2017). The offset may also be obtained from comparison with measurements from an absolute magnetometer or time-of-flight measurements of electrons emitted and observed by an electron drift instrument (Georgescu et al., 2006; Nakamura et al., 2014; Plaschke et al., 2014). Furthermore, the spin axis offset may also be obtained in regions of space where the fields are known, for instance, in diamagnetic cavities in the vicinity of comets (Goetz et al., 2016a, b).

From the preceding paragraphs, the reader might get the impression that in-flight calibration of magnetometers on spinning spacecraft is a solved issue; and in theory this is the case. However, as we will show in the following sections, the published methods for spin-aided calibration (Farrell et al., 1995; Kepko et al., 1996) are not optimal in practice because they do not take into account the physical behavior of the sensor package and the influence of a varying spacecraft attitude on the in-flight calibration.

This paper aims at identifying deficiencies and suggesting improvements with respect to the calibration Equations (1) and (2) and the specific choice of the calibration parameters. Thereafter, we identify optimal conditions for spin-related calibration parameter determination. Finally, we introduce advanced algorithms for parameter determination based on our findings, that lend itself for automation and distribution of calibration activities. A version of these algorithms is routinely applied to calibrate



**Table 1.** List of coordinate system notations used in this paper. After Table 1 in Kepko et al. (1996).

| notation | characteristics |
|----------|-----------------|
| S1, S2, S3 | spinning, non-orthogonal, sensor axes aligned |
| $Px, Py, Pz$ | spinning, orthogonal, sensor package system ($Pz$ = S3) |
| $x, y, z$ | spinning, orthogonal, spin-axis aligned ($z$-axis) |
| $X, Y, Z$ | non-spinning (inertial), orthogonal, spin-axis aligned ($Z = z$) |
| $X', Y', Z'$ | despun non-orthogonal coordinate system |

magnetometer data from the Magnetospheric Multiscale (MMS) mission (Burch et al., 2016; Torbert et al., 2016; Russell et al., 2016). The calibration principles and algorithms described here are based on developments at the Braunschweig University of Technology (Fornaçon et al., 2011) that have been successfully applied for decades to calibrate magnetometer data from, e.g., the Equator-S (Fornaçon et al., 1999), the Cluster (Balogh et al., 2001a, b), and the Time History of Events and Macroscale Interactions during Substorms (THEMIS) missions (Angelopoulos, 2008; Auster et al., 2008).

## 2   Calibration Equation and Parameters

Equations (1) and (2) in principle allow for any linear conversion of $\boldsymbol{B}_S$ into $\boldsymbol{B}$. The coupling matrix (2) is obviously split into two components:

$$\mathbf{C} \;=\; \boldsymbol{\Theta} \cdot \mathbf{G} \tag{3}$$

Here, the diagonal matrix $\mathbf{G}$ includes only the gains, and the matrix $\boldsymbol{\Theta}$ includes only the angular dependencies. Let's focus first on the matrix $\boldsymbol{\Theta}$. The parameters $\theta_1$, $\theta_2$, and $\theta_3$ are the angles between the three mutually non-orthogonal sensor axes (directions S1, S2, and S3) and the spin axis in $z$-direction in an orthogonal, spin axis aligned and spacecraft fixed coordinate system (directions $x$, $y$, and $z$). The parameters $\phi_1$, $\phi_2$, and $\phi_3$ correspond to the angles between the spacecraft fixed $x$-direction in the spin plane ($x$-$y$-plane, perpendicular to the spin axis) and the projections of the sensor axes onto that plane. For simplicity, the sensor axes S1, S2, and S3 are assumed to be approximately aligned with $x$ and $y$ and $z$. Note that all coordinate systems used in this paper are listed in Table 1.

The individual link of the sensor axes to a spacecraft fixed, spin axis aligned system is an issue here, as it does not reflect the actual situation on the spacecraft: There, the three sensor axes are typically packaged together into one sensor system. One of the design criteria of modern fluxgate magnetometer sensors is the temperature and long-term stability of the sensor axis directions as defined with respect to the sensor package. The angles between the sensor axes are usually well-known from ground calibration activities (e.g., Auster et al., 2008; Russell et al., 2016), and we can expect the three angles between the sensor axes to be relatively stable parameters. Consequently, in a first step, the magnetometer output in non-orthogonal sensor coordinates should be transformed into an orthogonal sensor package fixed coordinate system (coordinates: $Px, Py, Pz$, see



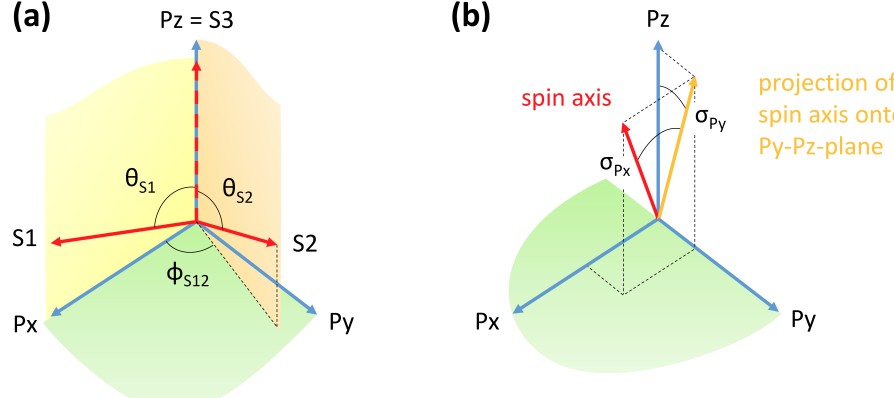

**Figure 1.** Sketch of the coordinate systems: (a) sensor axes in the sensor package coordinate system, (b) spin axis and rotation angles $\sigma_{\mathrm{P}x}$ and $\sigma_{\mathrm{P}y}$ in the sensor package coordinate system.

Table 1). The conversion matrix may have the following form:

$$
\mathbf{\Gamma} \;=\; \begin{pmatrix} \sin\theta_{\mathrm{S1}} & 0 & \cos\theta_{\mathrm{S1}} \\ \cos\phi_{\mathrm{S12}}\sin\theta_{\mathrm{S2}} & \sin\phi_{\mathrm{S12}}\sin\theta_{\mathrm{S2}} & \cos\theta_{\mathrm{S2}} \\ 0 & 0 & 1 \end{pmatrix}^{-1} \tag{4}
$$

Here, $\theta_{\mathrm{S1}}$ and $\theta_{\mathrm{S2}}$ are the angles between the sensor axes S1 and S2 with respect to S3=P$z$, and $\phi_{\mathrm{S12}}$ is the angle between the projections of S1 and S2 onto a plane perpendicular to S3, the P$x$-P$y$ plane. Note that S1 lies in the P$x$-P$z$ plane (see Figure 1a).

In the next step, the orientation of that sensor package system needs to be defined in a spacecraft-fixed spin axis aligned coordinate system. This latter transformation is expected to change every time there is a maneuver of the spacecraft, as fuel consumption will change the tensor of inertia and, thus, the spin axis direction in any spacecraft fixed coordinate system. The spin axis direction can be defined in the orthogonal sensor package system using two parameters or angles. During maneuvers, only those two parameters/angles should change, because the geometry inside the sensor package should not be affected. A rotation matrix $\mathbf{\Sigma}$ into a spin axis aligned coordinate system dependent on the two angles $\sigma_{\mathrm{P}x}$ and $\sigma_{\mathrm{P}y}$ can be defined as follows:

$$
\mathbf{\Sigma} \;=\; \begin{pmatrix} \cos\sigma_{\mathrm{P}x} & 0 & -\sin\sigma_{\mathrm{P}x} \\ 0 & 1 & 0 \\ \sin\sigma_{\mathrm{P}x} & 0 & \cos\sigma_{\mathrm{P}x} \end{pmatrix} \cdot \begin{pmatrix} 1 & 0 & 0 \\ 0 & \cos\sigma_{\mathrm{P}y} & -\sin\sigma_{\mathrm{P}y} \\ 0 & \sin\sigma_{\mathrm{P}y} & \cos\sigma_{\mathrm{P}y} \end{pmatrix} \tag{5}
$$

Here, $\sigma_{\mathrm{P}y}$ is the angle between P$z$ and the projection of the spin axis onto the P$y$-P$z$-plane, positive towards P$y$; $\sigma_{\mathrm{P}x}$ is the angle between that projection and the spin axis, positive towards P$x$. The angles are illustrated in Figure 1b. Note that the spin axis is assumed to be approximately aligned with the P$z =$ S3 axis. As a result, the angles $\sigma_{\mathrm{P}x}$ and $\sigma_{\mathrm{P}y}$ will be small and can be associated with the P$x$ and P$y$ coordinates of a unit vector that points in spin axis direction.



Using the angles $\sigma_{\mathrm{P}x}$ and $\sigma_{\mathrm{P}y}$ to define the spin axis direction is advantageous over using the angles $\theta_3$ and $\phi_3$, as the latter angle is badly defined if $\theta_3$ is small. Furthermore, it should also be noted that a change in direction of the spin axis requires an update of all angles of matrix $\boldsymbol{\Theta}$ as defined above, even though the magnetometer (sensor) itself is unaffected. Only two parameters ($\sigma_{\mathrm{P}x}$ and $\sigma_{\mathrm{P}y}$) need to be changed here to adapt the matrix $\boldsymbol{\Sigma}$ to the new spin axis direction.

To completely orient the sensor package (system) in the spin axis aligned coordinate system, a rotation about the spin axis (rotation matrix $\boldsymbol{\Phi}$) also needs to be taken into account:

$$\boldsymbol{\Phi} = \begin{pmatrix} \cos\phi_{\mathrm{a}} & -\sin\phi_{\mathrm{a}} & 0 \\ \sin\phi_{\mathrm{a}} & \cos\phi_{\mathrm{a}} & 0 \\ 0 & 0 & 1 \end{pmatrix} \tag{6}$$

As we will show later, this rotation does not affect the spin tone content in the despun magnetic field observations. The angle is affected by the orientation of a magnetometer boom and may change due to boom bending (Farrell et al., 1995).

Altogether, we can replace the orthogonalization and reorientation matrix $\boldsymbol{\Theta}$ by $\boldsymbol{\Phi} \cdot \boldsymbol{\Sigma} \cdot \boldsymbol{\Gamma}$ in Equation (3). Let's focus then again on the gain matrix $\mathbf{G}$ in that equation. As mentioned in the introduction, the spacecraft spin aids the determination of the ratio $g^2 = G_{\mathrm{S}1}/G_{\mathrm{S}2}$ of the spin plane gains, but not the absolute gains in the spin plane $G_{\mathrm{p}} = \sqrt{G_{\mathrm{S}1}G_{\mathrm{S}2}}$ and along the spin axis $G_{\mathrm{a}} = G_{\mathrm{S}3}$. Hence, it makes sense to use the parameters $g$ and $G_{\mathrm{p}}$ instead of $G_{\mathrm{S}1}$ and $G_{\mathrm{S}2}$ in the matrix $\mathbf{G}$, to decouple parameters that can be frequently updated from parameters that are only obtainable in flight from comparison to model fields

or measurements of other instruments:

$$\mathbf{G} = \begin{pmatrix} gG_{\mathrm{p}} & 0 & 0 \\ 0 & G_{\mathrm{p}}/g & 0 \\ 0 & 0 & G_{\mathrm{a}} \end{pmatrix} \tag{7}$$

Note that Kepko et al. (1996) use the difference of the inverse gains $\Delta G_{21} = 1/G_{\mathrm{S}2} - 1/G_{\mathrm{S}1}$ instead of $g$. However, later changes in the absolute gains $G_{\mathrm{S}1}$ and $G_{\mathrm{S}2}$ then require necessarily an update of $\Delta G_{21}$ in order to avoid perturbations at the second harmonic in the despun data. The gain ratio $g$, instead, is decoupled from changes in the absolute gains $G_{\mathrm{p}}$ and $G_{\mathrm{a}}$.

The gains should be stable parameters in the absence of temperature variations. These variations in the gains can be determined from ground calibration, resulting in a diagonal gain correction matrix $\mathbf{G}_{\mathrm{T}}(T_{\mathrm{s}}, T_{\mathrm{e}})$ that is dependent on the magnetometer sensor ($T_{\mathrm{s}}$) and electronics ($T_{\mathrm{e}}$) temperatures. That matrix should be directly applied to the magnetometer output $\boldsymbol{B}_{\mathrm{S}}$, requiring the knowledge of the sensor and electronics temperatures:

$$\boldsymbol{B}_{\mathrm{ST}} = \mathbf{G}_{\mathrm{T}}(T_{\mathrm{s}}, T_{\mathrm{e}}) \cdot \boldsymbol{B}_{\mathrm{S}} \tag{8}$$

The resulting temperature corrected output $\boldsymbol{B}_{\mathrm{ST}}$ may then be further converted to $\boldsymbol{B}$ via the coupling matrix $\mathbf{C} = \boldsymbol{\Phi} \cdot \boldsymbol{\Sigma} \cdot \boldsymbol{\Gamma} \cdot \mathbf{G}$ and the offset vector $\boldsymbol{O}$ using Equation (1), after replacing $\boldsymbol{B}_{\mathrm{S}}$ with $\boldsymbol{B}_{\mathrm{ST}}$. This also has the advantage that the further applied absolute gains $G_{\mathrm{p}}$ and $G_{\mathrm{a}}$ and the gain ratio $g^2$ should all be approximately 1 and unitless.

Altogether, we suggest to use the following improved calibration equation:

$$\boldsymbol{B} = \boldsymbol{\Phi} \cdot \boldsymbol{\Sigma} \cdot \boldsymbol{\Gamma} \cdot \mathbf{G} \cdot (\underbrace{\mathbf{G}_{\mathrm{T}}(T_{\mathrm{s}}, T_{\mathrm{e}}) \cdot \boldsymbol{B}_{\mathrm{S}}}_{=\boldsymbol{B}_{\mathrm{ST}}} - \boldsymbol{O}_{\mathrm{S}}) \tag{9}$$



with matrices defined in Equations (4) to (7) instead of the simpler Equations (1) and (2). The parameters whose determination is supported by the spacecraft spin are: $\theta_{S1}$, $\theta_{S2}$, $\phi_{S12}$, $\sigma_{Px}$, $\sigma_{Py}$, $g$, $O_{S1}$, and $O_{S2}$.

## 3 Calibration Parameter Influence on Spin Tone Harmonics

To determine the influence of the calibration parameters on the spin tone harmonic disturbances in the despun magnetic field measurements, we use a similar mathematical approach to Kepko et al. (1996) in this section. Based on the results, we go on to derive the optimal conditions for the determination of each parameter in section 4.

First, we compute the temperature corrected sensor output $\boldsymbol{B}_{ST}$ as a function of the external field $\boldsymbol{B}$ in the spinning coordinate system:

$$\boldsymbol{B}_{ST} = \mathbf{G}^{-1} \cdot \boldsymbol{\Gamma}^{-1} \cdot \boldsymbol{\Sigma}^{-1} \cdot \boldsymbol{\Phi}^{-1} \cdot \boldsymbol{B} + \boldsymbol{O}_{S} \tag{10}$$

We linearize all the matrices, using the following simplifying assumptions:

$$g \approx 1, \qquad G_{p} \approx 1, \qquad G_{a} \approx 1 \tag{11}$$

$$\sigma_{Px} \approx 0, \qquad \sigma_{Py} \approx 0 \tag{12}$$

$$\theta_{S1} \approx \pi/2, \quad \delta\theta_{S1} = \theta_{S1} - \pi/2 \approx 0 \tag{13}$$

$$\theta_{S2} \approx \pi/2, \quad \delta\theta_{S2} = \theta_{S2} - \pi/2 \approx 0 \tag{14}$$

$$\phi_{S12} \approx \pi/2, \quad \delta\phi_{S12} = \phi_{S12} - \pi/2 \approx 0 \tag{15}$$

Furthermore, we assume $\phi_a \approx 0$ without loss of generality. Dropping second order factors, we obtain the following linearized inverted matrices used in Equation (10):

$$\mathbf{G}^{-1} = \begin{pmatrix} 1/(gG_p) & 0 & 0 \\ 0 & g/G_p & 0 \\ 0 & 0 & 1/G_a \end{pmatrix} \tag{16}$$

$$\boldsymbol{\Gamma}^{-1} = \begin{pmatrix} 1 & 0 & -\delta\theta_{S1} \\ -\delta\phi_{S12} & 1 & -\delta\theta_{S2} \\ 0 & 0 & 1 \end{pmatrix} \tag{17}$$

$$\boldsymbol{\Sigma}^{-1} = \begin{pmatrix} 1 & 0 & \sigma_{Px} \\ 0 & 1 & \sigma_{Py} \\ -\sigma_{Px} & -\sigma_{Py} & 1 \end{pmatrix} \tag{18}$$

$$\boldsymbol{\Phi}^{-1} = \begin{pmatrix} 1 & \phi_a & 0 \\ -\phi_a & 1 & 0 \\ 0 & 0 & 1 \end{pmatrix} \tag{19}$$





Furthermore, without loss of generality, we assume the magnetic field in the despun (inertial) coordinate system (directions $X$, $Y$, and $Z$) to be in the $X$-$Z$-plane, and the spacecraft to spin around the $Z$-axis, which corresponds with the $z$-axis in the spacecraft fixed, spin aligned coordinate system (see Table 1). In that latter system, the field rotates and has the following form:

$$B_x = B_\mathrm{p}\cos\omega t = B_X\cos\omega t \tag{20}$$

$$B_y = B_\mathrm{p}\sin\omega t = B_X\sin\omega t \tag{21}$$

$$B_z = B_\mathrm{a} = B_Z \tag{22}$$

Inserting these relations in Equation (10) yields the expected temperature corrected output of the magnetometer in sensor coordinates. By applying the despin rotation matrix

$$\mathbf{D} = \begin{pmatrix} \cos\omega t & -\sin\omega t & 0 \\ \sin\omega t & \cos\omega t & 0 \\ 0 & 0 & 1 \end{pmatrix} \tag{23}$$

to Equation (10) to transform $\boldsymbol{B}_\mathrm{ST}$ into a non-orthogonal, despun coordinate system (directions $X'$, $Y'$, and $Z'$, see Table 1), after sorting by frequency and phase of the terms, and further dropping second order factors, we obtain the following relations.





They are structurally similar to Equations (11a), (11b), and (11c) in Kepko et al. (1996), but different in detail:

$$
\begin{aligned}
B_{X'} &= \frac{B_{\mathrm{p}}(1+g^2)}{2gG_{\mathrm{p}}} \\
&+ \cos\omega t\left[O_{\mathrm{S1}} + \frac{B_{\mathrm{a}}(\sigma_{\mathrm{P}x} - \delta\theta_{\mathrm{S1}})}{gG_{\mathrm{p}}}\right] \\
&- \sin\omega t\left[O_{\mathrm{S2}} + \frac{gB_{\mathrm{a}}(\sigma_{\mathrm{P}y} - \delta\theta_{\mathrm{S2}})}{G_{\mathrm{p}}}\right] \\
&+ \cos 2\omega t\left[\frac{B_{\mathrm{p}}(1-g^2)}{2gG_{\mathrm{p}}}\right] \\
&+ \sin 2\omega t\frac{B_{\mathrm{p}}}{2G_{\mathrm{p}}}\left[g\phi_{\mathrm{a}} - \frac{\phi_{\mathrm{a}}}{g} + g\delta\phi_{\mathrm{S12}}\right]
\end{aligned}
\tag{24}
$$

$$
\begin{aligned}
B_{Y'} &= -\frac{B_{\mathrm{p}}}{2G_{\mathrm{p}}}\left[\frac{1+g^2}{g}\phi_{\mathrm{a}} + g\delta\phi_{\mathrm{S12}}\right] \\
&+ \cos\omega t\left[O_{\mathrm{S2}} + \frac{gB_{\mathrm{a}}(\sigma_{\mathrm{P}y} - \delta\theta_{\mathrm{S2}})}{G_{\mathrm{p}}}\right] \\
&+ \sin\omega t\left[O_{\mathrm{S1}} + \frac{B_{\mathrm{a}}(\sigma_{\mathrm{P}x} - \delta\theta_{\mathrm{S1}})}{gG_{\mathrm{p}}}\right] \\
&- \cos 2\omega t\frac{B_{\mathrm{p}}}{2G_{\mathrm{p}}}\left[g\phi_{\mathrm{a}} - \frac{\phi_{\mathrm{a}}}{g} + g\delta\phi_{\mathrm{S12}}\right] \\
&+ \sin 2\omega t\left[\frac{B_{\mathrm{p}}(1-g^2)}{2gG_{\mathrm{p}}}\right]
\end{aligned}
\tag{25}
$$

$$
\begin{aligned}
B_{Z'} &= \frac{B_{\mathrm{a}}}{G_{\mathrm{a}}} + O_{\mathrm{S3}} \\
&- \cos\omega t\frac{B_{\mathrm{p}}\sigma_{\mathrm{P}x}}{G_{\mathrm{a}}} \\
&+ \sin\omega t\frac{B_{\mathrm{p}}\sigma_{\mathrm{P}y}}{G_{\mathrm{a}}}
\end{aligned}
\tag{26}
$$

These equations show how the parameters affect the signal content at the spin tone harmonics in the despun measurements.

## 4 Favorable Conditions for the Determination of the Calibration Parameters

From the factors pertaining to the first and second harmonic terms of $B_{X'}$, $B_{Y'}$, and $B_{Z'}$ (Equations 24 to 26) it is possible to derive the conditions that should be favorable for the determination of the 8 previously mentioned parameters. These factors are:

$$
\left[O_{\mathrm{S1}} + \frac{B_{\mathrm{a}}(\sigma_{\mathrm{P}x} - \delta\theta_{\mathrm{S1}})}{gG_{\mathrm{p}}}\right] \quad \text{and} \quad \left[O_{\mathrm{S2}} + \frac{gB_{\mathrm{a}}(\sigma_{\mathrm{P}y} - \delta\theta_{\mathrm{S2}})}{G_{\mathrm{p}}}\right]
\tag{27}
$$

$$
\left[\frac{B_{\mathrm{p}}\sigma_{\mathrm{P}x}}{G_{\mathrm{a}}}\right] \quad \text{and} \quad \left[\frac{B_{\mathrm{p}}\sigma_{\mathrm{P}y}}{G_{\mathrm{a}}}\right]
\tag{28}
$$

$$
\left[\frac{B_{\mathrm{p}}(1-g^2)}{2gG_{\mathrm{p}}}\right] \quad \text{and} \quad \frac{B_{\mathrm{p}}}{2G_{\mathrm{p}}}\left[g\phi_{\mathrm{a}} - \frac{\phi_{\mathrm{a}}}{g} + g\delta\phi_{\mathrm{S12}}\right]
\tag{29}
$$



**Table 2.** Parameters and favorable conditions

| Group | Parameters | Disturbances | Conditions | Uncertainties |
|---|---|---|---|---|
| 1 | $\sigma_{\mathrm{P}x}$ and $\sigma_{\mathrm{P}y}$ | at $\omega$ along spin axis | high $B_{\mathrm{p}}$, low $F_{\mathrm{a}}$ | $\Delta\sigma_{\mathrm{P}x/y} \approx F_{\mathrm{a}}/B_{\mathrm{p}}$ |
| 2 | $g$ and $\delta\phi_{\mathrm{S}12}$ | at $2\omega$ in spin plane | high $B_{\mathrm{p}}$, low $F_{2\mathrm{p}}$ | $\Delta g \approx F_{2\mathrm{p}}/B_{\mathrm{p}}$ and $\Delta\phi_{\mathrm{S}12} \approx 2F_{2\mathrm{p}}/B_{\mathrm{p}}$ |
| 3 | $O_{\mathrm{S}1}$ and $O_{\mathrm{S}1}$ | at $\omega$ in spin plane | low $B_{\mathrm{a}}$, low $F_{\mathrm{p}}$ | $\Delta O_{\mathrm{S}1/2} \approx F_{\mathrm{p}} + B_{\mathrm{a}}\Delta\sigma_{\mathrm{P}x/y} + B_{\mathrm{a}}\Delta\theta_{\mathrm{S}1/2}$ |
| 4 | $\delta\theta_{\mathrm{S}1}$ and $\delta\theta_{\mathrm{S}2}$ | at $\omega$ in spin plane | high $B_{\mathrm{a}}$, low $F_{\mathrm{p}}$ | $\Delta\theta_{\mathrm{S}1/2} \approx F_{\mathrm{p}}/B_{\mathrm{a}} + \Delta O_{\mathrm{S}1/2}/B_{\mathrm{a}} + \Delta\sigma_{\mathrm{P}x/y}$ |

Here, the factors (27) and (28) pertain to the spin tone disturbances in the despun spin plane and spin axis components, respectively, and (29) pertains to the second harmonic frequency disturbance (double spin tone frequency) in the spin plane components.

As can be seen, the first factor of the latter group (29) is dependent on $B_{\mathrm{p}}$, the external field in the spin plane which we assume to be constant, on $G_{\mathrm{p}}$, the absolute gain factor in the spin plane which should be approximately 1, and on $1/g - g$, which is 0 only if $g = 1$. Hence, the presence of one part of the second harmonic disturbance, though modulated by $B_{\mathrm{p}}$, is ultimately dependent only on $g$, the ratio of spin plane gains. Consequently, this relation can be used to determine $g$ correctly. The signal to do that and, in particular, the signal to noise ratio (SNR) is larger if $B_{\mathrm{p}}$ is larger. We capture this relation in the second line of Table 2. As the second harmonic disturbance in the spin plane is to be minimized to get $g$, the natural fluctuations around that frequency (of amplitude $F_{2\mathrm{p}}$) should also be low in the spin plane. The uncertainty in $g$ is then expected to be on the order of $F_{2\mathrm{p}}/B_{\mathrm{p}}$.

The same is true for the complementary factor, on the right side of (29): Also this second harmonic disturbance is modulated by $B_{\mathrm{p}}$. When $g$ is accurately determined, then the $\phi_{\mathrm{a}}$ influence vanishes, and the entire factor can only vanish by correctly choosing $\delta\phi_{\mathrm{S}12}$. Hence, to determine this parameter accurately, also $B_{\mathrm{p}}$ should be large and the natural fluctuations at the second harmonic should be of low amplitude (low $F_{2\mathrm{p}}$). The uncertainty $\Delta\phi_{\mathrm{S}12}$ of $\delta\phi_{\mathrm{S}12}$ and, ultimately, $\phi_{\mathrm{S}12}$ is expected to be on the order of $2F_{2\mathrm{p}}/B_{\mathrm{p}}$ (see line 2 in Table 2).

Let's focus on the group of factors (28). The spin frequency disturbance is clearly modulated by $B_{\mathrm{p}}$, as $G_{\mathrm{a}}$ should be close to 1, so $B_{\mathrm{p}}$ benefits the SNR. These disturbances vanish if $\sigma_{\mathrm{P}x}$ and $\sigma_{\mathrm{P}y}$ become 0, i.e., if they are precisely determined. A low amplitude in the natural fluctuations at the spin frequency along the spin axis $F_{\mathrm{a}}$ would also support the determination. The uncertainty in $\sigma_{\mathrm{P}x}$ and $\sigma_{\mathrm{P}y}$ is then expected to be on the order of $\Delta\sigma_{\mathrm{P}x/y} \approx F_{\mathrm{a}}/B_{\mathrm{p}}$ (line 1 in Table 2).

The first set of factors in (27) pertain to the spin frequency disturbances in the spin plane components. They consist of two parts: a spin plane offset component $O_{\mathrm{S}1}$ or $O_{\mathrm{S}2}$, and a term that is modulated by $B_{\mathrm{a}}$ and which may vanish if the difference $(\sigma_{\mathrm{P}x} - \delta\theta_{\mathrm{S}1})$ or $(\sigma_{\mathrm{P}y} - \delta\theta_{\mathrm{S}2})$ vanishes. Obviously, if $B_{\mathrm{a}}$ vanishes, then the spin plane spin frequency disturbances can only come from the spin plane offset components. Hence, for their determination it is beneficial if the spin axis field $B_{\mathrm{a}}$ is low and if the natural fluctuation level around the spin frequency in the spin plane $F_{\mathrm{p}}$ is low. The uncertainty in $O_{\mathrm{S}1}$ and $O_{\mathrm{S}2}$ is then expected to be on the order of $F_{\mathrm{p}} + B_{\mathrm{a}}\Delta\sigma_{\mathrm{P}x/y} + B_{\mathrm{a}}\Delta\theta_{\mathrm{S}1/2}$ (see line 3 of Table 2).





The remaining elevation angles $\delta\theta_{S1}$ and $\delta\theta_{S2}$ are most difficult to determine: it is beneficial if the spin axis field $B_a$ is high. In addition, however, it is necessary that the spin axis itself is well determined, as the parameters $\sigma_{Px}$ and $\sigma_{Py}$ equally influence the spin tone signal in the spin plane as $\delta\theta_{S1}$ and $\delta\theta_{S2}$. Note that $\sigma_{Px}$ and $\sigma_{Py}$ can be independently determined by minimizing the spin frequency disturbances in the spin axis component. $F_p$ should again be low. Altogether, the uncertainty in

$\delta\theta_{S1/2}$ is on the order of $\Delta\theta_{S1/2} \approx F_p/B_a + \Delta O_{S1/2}/B_a + \Delta\sigma_{Px/y}$ (see line 4 of Table 2).

## 5 Parameter Determination

Based on the findings from the previous section, we propose algorithms to determine the 8 spin-related parameters in an iterative manner (sections 5.2 to 5.5). The algorithms are based on computing estimates of the parameters for short intervals, and evaluate the uncertainties of those estimates based on the uncertainties indicated in Table 2. Then, the estimates with

uncertainties below a certain acceptable threshold are chosen to form the basis of one parameter correction.

### 5.1 Precalibration

The temperature dependent gains $\mathbf{G}_T(T_s, T_s)$ determined on ground should be used to convert the raw magnetometer output $\boldsymbol{B}_S$ to a precalibrated, temperature corrected intermediate product $\boldsymbol{B}_{ST}$, according to Equation (8).

The offset vector $\boldsymbol{O}_S$ and the calibration matrices $\boldsymbol{\Phi}$, $\boldsymbol{\Sigma}$, $\boldsymbol{\Gamma}$, and $\mathbf{G}$ should be initiated with the best known values at the time

of calibration. At the beginning, these will be ground obtained values:

– for $\theta_{S1}$, $\theta_{S2}$, $\phi_{S12}$, and $\boldsymbol{O}_S$ from ground magnetometer calibration,

– for $\phi_a$ from nominal spacecraft design or mirror/laser based alignment measurements,

– for $\sigma_{Px}$ and $\sigma_{Py}$ from an initial estimate of the spin axis direction (alternatively $\sigma_{Px} = \sigma_{Py} = 0$ may be chosen),

– and $G_p = G_a = g = 1$ due to precalibration.

If in-flight calibration has already taken place, then these values will be superseded by better in-flight determined values.

### 5.2 Calibration of the Spin Axis Direction

The entire interval of magnetic field measurements should be divided into small (overlapping) subintervals of length $t_{int} = 2\pi n/\omega$, with $n \in \mathbb{N}$. The factor $n$ should not be too small; hence, the subintervals should contain a number of spin periods, so that the spin tone at the spin frequency and also the power around that frequency can be accurately determined. On the other

hand, subintervals should not be too large, so that the field/environmental conditions can be assumed constant.

For each of the subintervals, the uncertainties $\Delta\sigma_{Px/y} \approx F_a/B_p$ need to be calculated (line 1 in Table 2). Conservatively, we choose $B_p$ to be the minimal modulus of the spin plane field over the subinterval: $\min\left(\sqrt{B_x^2 + B_y^2}\right)$. $F_a$ can be estimated by taking the maximum of the discrete Fourier components $F_{a\pm}$ of the spin axis magnetic field $B_z$ at frequencies $\omega_\pm$ that are



slightly over and under the spin frequency: $\omega_\pm = 2\pi n_\pm/t_{\text{int}}$ with $n_\pm \in \mathbb{N}$ and slightly over/under $n$:

$$F_{\text{a}\pm} = \mathcal{F}(B_z, \omega_\pm) = \left| \frac{2}{N} \sum_{k=0}^{N-1} B_z(t_0 + k\delta t) \exp(-i\omega_\pm k\delta t) \right| \qquad (30)$$

$$F_{\text{a}} = \max(F_{\text{a}\pm}) \qquad (31)$$

Here $t_0$ is the start of a subinterval considered, $N$ is the number of magnetic field measurement samples in that subinterval,

and $\delta t$ is the sampling period.

From here on, we use $\mathcal{F}(B,\omega)$ to denote the Fourier component of $B$ at frequency $\omega$. It should be noted that it may be recommended to de-trend the $B$ data before computing $\mathcal{F}(B,\omega)$. Linear trends will not occur if the external field can be assumed to be constant. In many real applications, however, the spacecraft will move through field gradients during subintervals considered, and in these cases, the linear trend in the field measurements will increase the spectral content across the spectrum.

Parameter estimates $\sigma_{\text{P}x}$ and $\sigma_{\text{P}y}$ are determined by minimization of the spin tone $S_{\text{a}}$ in the spin axis component: $S_{\text{a}} = \mathcal{F}(B_z,\omega)$. This minimization is performed for each subinterval. Hence, we obtain for each subinterval one estimate for $\sigma_{\text{P}x}$, for $\sigma_{\text{P}y}$, and for the uncertainty $\Delta\sigma_{\text{P}x/y}$. A final parameter update for $\sigma_{\text{P}x}$ and $\sigma_{\text{P}y}$ for the entire interval of interest may be obtained by selecting the most accurate subinterval estimates of those parameters, pertaining to minimal uncertainties $\Delta\sigma_{\text{P}x/y}$. From those estimates, the median or average may be computed. The selection of the best estimates can be threshold based with

respect to $\Delta\sigma_{\text{P}x/y}$.

## 5.3 Calibration of Gain Ratio and Azimuthal Angle

As detailed in the previous section 5.2, an interval of interest is divided into short (overlapping) subintervals of length $t_{\text{int}} = 2\pi n/\omega$. For each of these subintervals, the uncertainties $\Delta g$ and $\Delta\phi_{\text{S12}}$ are computed (see line 2 in Table 2). Therefor, the fluctuation amplitudes $F_{\text{2p}} = \max(\mathcal{F}(\sqrt{B_x^2 + B_y^2}, 2\omega_\pm))$ need to be computed, with $2\omega_\pm = 4\pi n_\pm/t_{\text{int}}$ and $n_\pm \in \mathbb{N}$

slightly over/under $n$. Subsequently, the parameters $g$ and $\delta\phi_{\text{S12}}$ are determined for each subinterval by minimization of $S_{\text{2p}} = \mathcal{F}(\sqrt{B_x^2 + B_y^2}, 2\omega)$. From the set of $g$ and $\delta\phi_{\text{S12}}$ estimates from all subintervals, those associated to lowest uncertainties can be chosen to yield final updates for $g$ and $\delta\phi_{\text{S12}}$.

It should be noted that we are using here the modulus of the spin plane field ($\sqrt{B_x^2 + B_y^2} = \sqrt{B_X^2 + B_Y^2}$) to compute $F_{\text{2p}}$ and $S_{\text{2p}}$ instead of any individual spin plane component ($B_X$ or $B_Y$) in a despun coordinate system, as would be suggested

by the analytical treatment outlined in section 3. Both approaches (using the modulus or a despun component) are, however, mathematically equivalent. To show this, we can compute $\sqrt{B_{X'}^2 + B_{Y'}^2}$ using Equations (24) and (25). From the sum $B_{X'}^2 + B_{Y'}^2$, only those terms are large which contain the first term of Equation (24), because all other terms are products of multiple small factors and, hence, can be omitted due to linearization. Taking that into account, we obtain $\sqrt{B_{X'}^2 + B_{Y'}^2} \approx B_{X'}$. Hence, $\sqrt{B_x^2 + B_y^2}$ contains field and variations corresponding to the despun component, along which the external field is pointing.

Evaluating $\sqrt{B_x^2 + B_y^2}$ is hence equivalent to evaluating $B_X$ if the field points in the $X$-direction. This result is based on the assumption of the spin tone and second harmonic terms being small in comparison to the constant spin plane magnetic field, which should be fulfilled even in low field conditions if the initial set of calibration parameters is not too inaccurate.



Note also that it is not possible to obtain additional information with respect to the calibration parameters by evaluating the field-perpendicular component $B_Y$, because the coefficients pertaining to the $\sin$ and $\cos$ terms of $B_{X'}$ and $B_{Y'}$ are the same (compare Equations 24 and 25).

The equivalence of the approaches (using the modulus or a despun component) brings up two questions: (i) Why did we not use the modulus when calculating the influences of the spin-related parameters in section 3 and (ii) why would we prefer using the modulus over a despun component here and in any practical application of the calibration algorithms outlined in this section 5? The answer to question (i) is that the mathematical treatment of the modulus is slightly more involved than the treatment of the individual despun coordinates. Furthermore, section 3 follows the approach of Kepko et al. (1996) who also use despun coordinates. Thereby, our results from section 3 become directly comparable to the results of their study. In their and our analytical treatments, the despinning process is exactly defined, perfectly known, and accurate. Hence, it does not introduce additional uncertainty into the calibration process. This latter statement is not true in any real application, which leads us to the answer of question (ii): The modulus of the spin plane field is readily available in any spinning coordinate system. Despinning is not necessary for magnetometer calibration, and it is not advised because it could introduce additional, unnecessary uncertainty.

## 5.4 Calibration of Spin Plane Offsets

The uncertainties for each subinterval are computed as suggested in line 3 of Table 2. Thereto, the maximum of the spin axis field ($B_a = \max|B_z|$) over each subinterval should be used. $F_p$ is evaluated following Equation (31). Furthermore, estimates for the uncertainties $\Delta\sigma_{Px/y}$ and $\Delta\theta_{S1/2}$ need to be obtained. These may be based on the variability of the selected estimates of $\sigma_{Px/y}$ (see section 5.2) and $\delta\theta_{S1/2}$ (see next section 5.5) used to compute the final values of those parameters.

The offset estimates $O_{S1}$ and $O_{S2}$ are determined for each subinterval by minimization of $S_p = \mathcal{F}(\sqrt{B_x^2 + B_y^2}, \omega)$. From the set of $O_{S1}$ and $O_{S2}$ estimates, the most accurate can be chosen to compute final updates for the spin plane offsets. It should be noted that the offsets are known to be the most variable parameters. Hence, it could be desirable to compute final offset updates more often than updates of other spin-related parameters, if possible.

## 5.5 Calibration of Elevation Angles

The uncertainties for each subinterval are computed as suggested in line 4 of Table 2, this time using $B_a = \min|B_z|$. Estimates for the uncertainties $\Delta\sigma_{Px/y}$ and $\Delta O_{S1/2}$ need to be obtained, e.g., from the variability of selected $\sigma_{Px/y}$ and $O_{S1/2}$ estimates. Subsequently, the elevation angles $\delta\theta_{S1}$ and $\delta\theta_{S2}$ are determined for each subinterval by minimization of $S_p = \mathcal{F}(\sqrt{B_x^2 + B_y^2}, \omega)$. From the set of $\delta\theta_{S1}$ and $\delta\theta_{S2}$ estimates, the most accurate can be chosen (lowest uncertainties) to yield final updates of those parameters.

It should be noted that the same quantity $S_p$ is minimized to obtain the elevation angles $\delta\theta_{S1}$ and $\delta\theta_{S2}$ and the offset components $O_{S1}$ and $O_{S2}$. Hence, the final selection of estimates according to the uncertainties $\Delta\theta_{S1/2}$ and $\Delta O_{S1/2}$, which are heavily dependent on $|B_z|$, is very important here. In low $|B_z|$, minimization of $S_p$ yields the offset components, whereas in



high fields the offsets do not matter any more and any spin tone may safely be attributed to an incorrect choice of the elevation angles, if the spin axis direction is precisely known.

## 5.6 Exclusion of Data Intervals

Certain intervals may be excluded from parameter determination, as some of the underlying assumptions may not be met well. For instance, intervals featuring large spacecraft/sensor temperature changes should be avoided, as parameters may vary within such intervals. Hence, uncertainties in the parameters may be significantly higher than what is reflected in the uncertainty estimates stated in Table 2. Large temperature variations are expected during eclipse intervals, where the spacecraft is in shadow (e.g., of Earth), and hours after eclipse intervals as temperatures relax to stationary values. Furthermore, magnetic field measurements at saturation levels need to be avoided. Lastly, intervals during and after spacecraft maneuvers may be problematic for calibration, as the spin axis will fluctuate during maneuvers and nutation may be visible for periods of time after maneuvers. It should be noted that all these considerations are spacecraft and orbit specific.

## 6 Application to THEMIS Data

To ascertain the accuracies that parameters may be determined with in different regions of near Earth space, on a highly elliptical orbit around Earth, we apply the algorithms detailed above to two days (20 and 21 July 2007) of THEMIS-C (Angelopoulos, 2008) fluxgate magnetometer (FGM) data (Auster et al., 2008). The data are available at $4\,\mathrm{Hz}$ sampling frequency (data product: FGL); they are already fully calibrated and the applied calibration parameters do not change over the two days considered. The magnitudes of the magnetic field along the spin axis $|B_z|$ and in the spin plane $\sqrt{B_x^2 + B_y^2}$ are displayed in Figure 2a in red and blue, respectively.

The different regions that THEMIS-C went through during these two particular days are best identified using the omni-directional ion spectral energy flux densities, measured by the electrostatic analyzer (ESA, McFadden et al., 2008) and displayed in Figure 2b. At the beginning of 20 July 2007, THEMIC-C is located in the dayside magnetosheath. This is clearly visible in the broad ion energy spectrum which is characteristic for the thermalized solar wind plasma population present downstream of the bow shock. THEMIS-C fully transitions through the magnetopause into the magnetosphere at about 05:06 UT, moving inbound towards perigee at about 10:27 UT. At about 15:33 UT, THEMIS-C went back into the magnetosheath until about 22:33 UT, when it transitioned through the bow shock into the solar wind, characterized by a narrow energy signature corresponding to a cold plasma moving at the solar wind speed. On 21 July, THEMIS-C went back into the magnetosheath at about 06:07 UT, and then went further into the magnetosphere at about 10:53 UT. The perigee pass at that day took place at about 17:47 UT.

As can be seen in panel (a), the solar wind interval is characterized by low magnetic fields, typically below $10\,\mathrm{nT}$. In the dayside magnetosheath, the field strength is somewhat higher, on the order of a few tens of $\mathrm{nT}$, and highly fluctuating. Inside the magnetosphere, the fluctuation level is again low. The lowest field strengths of a few tens of $\mathrm{nT}$ are measured on the earthward side of the magnetopause, so just inside the inner magnetosphere. The field strength continuously increases towards

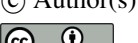



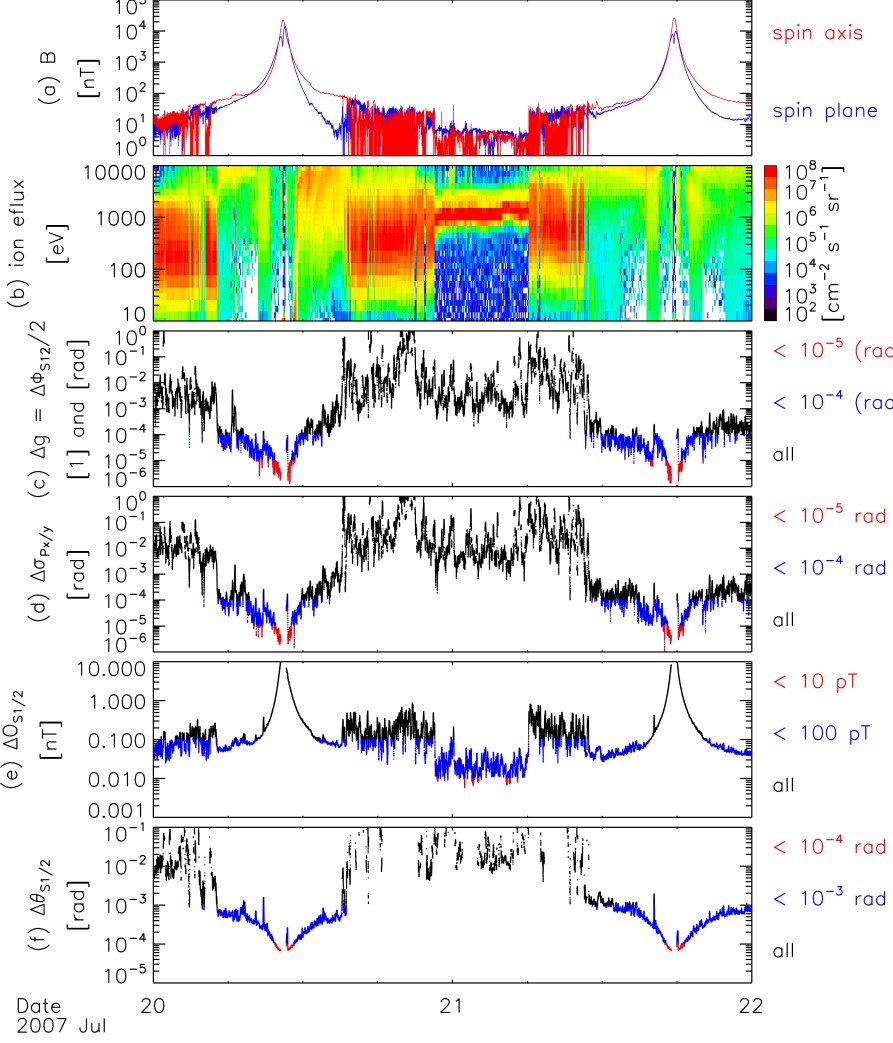

**Figure 2.** From top to bottom: (a) magnitude of the spin axis and spin plane magnetic fields in red and blue, (b) omni-directional ion spectral energy flux densities, (c) – (f) uncertainties of the estimates of the respective calibration parameters calculated in accordance to Table 2.

Earth. On this particular THEMIS-C orbit, field strengths on the order of $10^4$ nT are reached along the spin axis and in the spin plane close to perigee. As discussed above, both the fluctuation levels and field strengths have a major influence on the expected uncertainties of the calibration parameter estimates. We determine the parameters and the corresponding uncertainties for overlapping subintervals of 100 spin periods each, a spin period lasting for approximately 3 s. Hence, the subintervals have interval lengths of approximately 5 minutes. Note that we do not consider subintervals containing fields above $2 \cdot 10^4$ nT, due to FGM instrument saturation, and also excluded intervals in eclipse (Earth shadow) around perigee that lasted for approximately 22 min per orbit.





Subinterval lengths of 100 spin periods ensure good estimates of the power at around (and double) the spin frequency $\omega = 2\pi/(3\,\mathrm{s}) \approx 2\,\mathrm{rad/s}$, while the calibration parameters to be determined and the ambient magnetic field conditions may well be considered constant over such short intervals. Estimates of the power $F_{\mathrm{a}\pm}$, $F_{\mathrm{p}\pm}$, and $F_{2\mathrm{p}\pm}$ around (double) the spin frequency are taken at $85\%$ and $115\%$ of $\omega$ and at $185\%$ and $215\%$ of $\omega$, respectively. Following the equations from Table 2 and from the subsections 5.2 to 5.5 above, we determine the uncertainties for the calibration parameter estimates for all subintervals. They are shown in Figures 2c to 2f.

Figures 2c and 2d show the uncertainties $\Delta g = \Delta\phi_{\mathrm{S}12}/2$ and $\Delta\sigma_{\mathrm{P}x/y}$. For the corresponding parameters, uncertainties on the order of $10^{-4}\,(\mathrm{rad})$ are generally acceptable. In the case of the gain ratio parameter $g$, an error of $10^{-4}$ would translate into an absolute error of $1\,\mathrm{nT}$ in $10000\,\mathrm{nT}$ fields. With respect to the angle $\phi_{\mathrm{S}12}$ (or $\delta\phi_{\mathrm{S}12}$) and to the spin axis angles $\sigma_{\mathrm{P}x/y}$, an error of $1 \cdot 10^{-4}\,\mathrm{rad}$ is equivalent to approximately $0.5\%$ of a degree. Uncertainties below $10^{-4}\,(\mathrm{rad})$ are marked in blue in Figures 2c and 2d. As can be seen, estimates of $g$, $\phi_{\mathrm{S}12}$, and $\sigma_{\mathrm{P}x/y}$ with uncertainties below this threshold can be obtained almost everywhere in the inner magnetosphere, where fields are relatively stable, but not in the magnetosheath (fluctuations too high) or in the solar wind (fields too low). Estimates associated with uncertainties below $10^{-5}\,(\mathrm{rad})$ are marked in red in Figures 2c and 2d. These are obtained only in the regions of highest ambient fields, close to perigee.

The parameter estimates themselves ($g$, $\delta\phi_{\mathrm{S}12}$, $\sigma_{\mathrm{P}x}$, and $\sigma_{\mathrm{P}y}$) are shown in Figures 3a to 3d as a function of their respective uncertainties. Again, uncertainty thresholds of $10^{-4}\,(\mathrm{rad})$ and $10^{-5}\,(\mathrm{rad})$ are marked in blue and red, respectively. Taking the averages of the estimates associated with uncertainties below $10^{-5}\,(\mathrm{rad})$, we obtain:

$$\langle g \rangle = 0.99998 \pm 0.00004 \tag{32}$$

$$\langle \delta\phi_{\mathrm{S}12} \rangle = (-3 \pm 6) \cdot 10^{-5}\,\mathrm{rad} \tag{33}$$

$$\langle \sigma_{\mathrm{P}x} \rangle = (-7 \pm 6) \cdot 10^{-5}\,\mathrm{rad} \tag{34}$$

$$\langle \sigma_{\mathrm{P}y} \rangle = (-1.7 \pm 0.4) \cdot 10^{-4}\,\mathrm{rad} \tag{35}$$

Here, the error values are the corresponding standard deviations of the estimates. We see that all parameters are close to 0 (or 1 in the case of $g$); an update may only be advised for $\sigma_{\mathrm{P}y}$, as its deviation from 0 is significantly larger than the error value (see Figure 3d).

In Figures 3c and 3d, a split in values associated to low uncertainties can be clearly seen. A closer look on this phenomenon reveals that lower/higher $\sigma_{\mathrm{P}x}/\sigma_{\mathrm{P}y}$ values correspond to times before/after perigee passes. Hence, the spin axis direction in the orthogonalized sensor package coordinate system changes during perigee. This might be related to a temperature driven change in spacecraft geometry, i.e., in boom alignment to the spacecraft body, occurring in eclipse during perigee passes.

In order to calculate the uncertainties of the offset and elevation angle estimates ($\Delta O_{\mathrm{S}1/2}$ and $\Delta\theta_{\mathrm{S}1/2}$, see lines 3 and 4 in Table 2), we have to assume uncertainties in the knowledge of the spin axis direction angles ($\Delta\sigma_{\mathrm{P}x/y}$), the offsets ($\Delta O_{\mathrm{S}1/2}$), and the elevation angles ($\Delta\theta_{\mathrm{S}1/2}$). Based on Equation (35), we set $\Delta\sigma_{\mathrm{P}x/y} = 6 \cdot 10^{-5}\,\mathrm{rad}$. Furthermore, as we can justify a posteriori based on Equations (37) and (39), we set $\Delta O_{\mathrm{S}1/2} = 25\,\mathrm{pT}$ and $\Delta\theta_{\mathrm{S}1/2} = 7 \cdot 10^{-4}\,\mathrm{rad}$. Therewith, we obtain the uncertainty estimates per subinterval shown in Figures 2e and 2f.





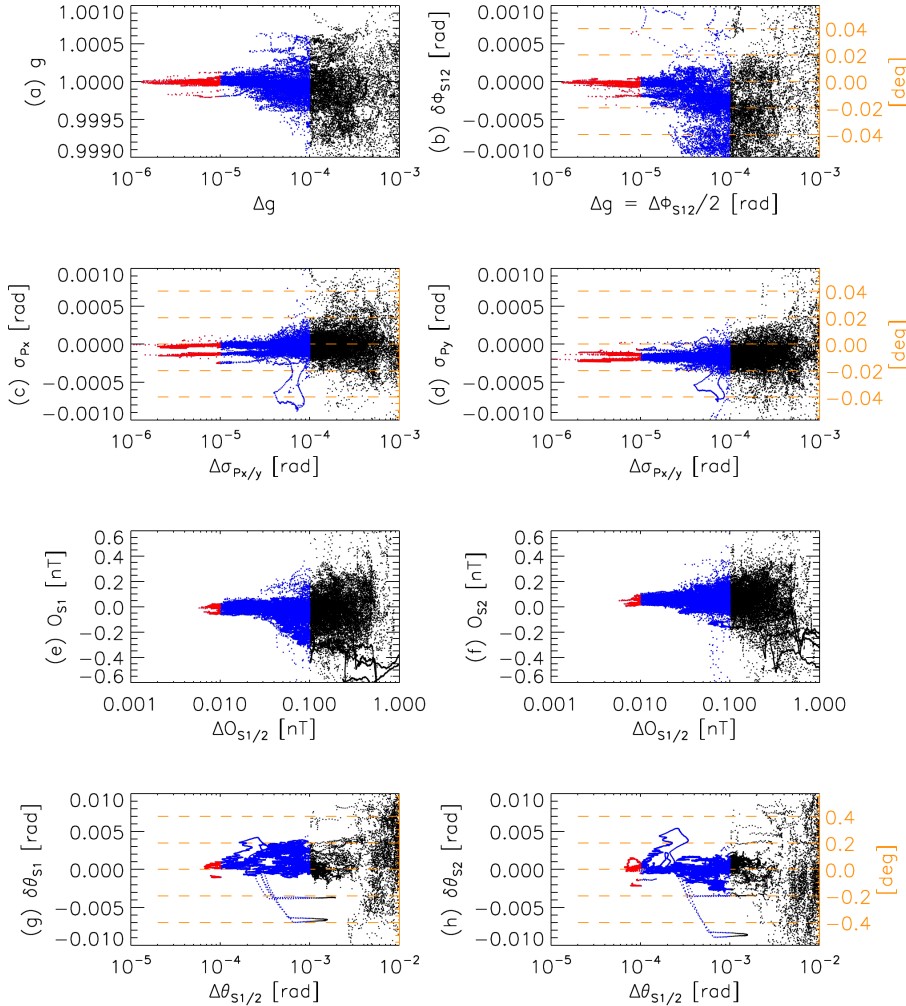

**Figure 3.** Calibration parameter estimates as a function of their respective uncertainties. Threshold levels for blue and red marked estimates are the same as in Figure 2. Panels (b) to (d), (g), and (h) have secondary axes in orange, showing parameter values in degrees.

The offsets directly influence the absolute accuracies of the magnetic field measurements. Typically, uncertainties on the order or below $0.1\,\text{nT}$ are desired in low fields. Uncertainties meeting this threshold are marked in blue in Figure 2e. As can be seen, corresponding offset estimates can be routinely obtained in the solar wind, due to the low fields, and also in the outer, low field parts of the inner magnetosphere. Within the magnetosheath, however, many estimate uncertainties surpass the threshold as the fluctuations levels are too high for accurate offset determinations. Estimates with uncertainties below $10\,\text{pT}$ (in red) can only be obtained in the solar wind at low fields. From those (red dots in Figures 3e and 3f), we obtain average offsets of:

$$\langle O_{\text{S1}} \rangle = (-0.007 \pm 0.023)\,\text{nT} \tag{36}$$

$$\langle O_{\text{S2}} \rangle = (0.036 \pm 0.025)\,\text{nT} \tag{37}$$



The error values here motivate the choice of $\Delta O_{\mathrm{S1/2}}$ for the computation of the uncertainties of the elevation angles. These angles should also be known to the order of $10^{-4}\,\mathrm{rad}$. Unfortunately, estimates with uncertainties lower than this threshold are only obtained in very high fields, close to perigee, as can be seen by the red dots in Figures 2f, 3g, and 3h. The blue dots correspond to the lower threshold of $10^{-3}\,\mathrm{rad}$ in this case, already equivalent to $5.7\%$ of a degree in angular uncertainty. From the $\delta\theta_{\mathrm{S1/2}}$ estimates pertaining to uncertainties lower than $10^{-4}\,\mathrm{rad}$ we obtain the following averages:

$$\langle\delta\theta_{\mathrm{S1}}\rangle \quad = \quad (5\pm 4)\cdot 10^{-4}\,\mathrm{rad} \tag{38}$$

$$\langle\delta\theta_{\mathrm{S2}}\rangle \quad = \quad (3\pm 7)\cdot 10^{-4}\,\mathrm{rad} \tag{39}$$

Within the group of sensor orthogonalization angles ($\delta\theta_{\mathrm{S1}}$, $\delta\theta_{\mathrm{S2}}$, and $\delta\phi_{\mathrm{S12}}$) and spin axis angles ($\sigma_{\mathrm{P}x}$ and $\sigma_{\mathrm{P}y}$), these elevation angles are least accurately defined. Apparently, it is difficult to determine them to accuracies on the order of $10^{-4}\,\mathrm{rad}$ or better. When determined on ground, in higher fields, the parameters $\delta\theta_{\mathrm{S1/2}}$ may however be better determined, with lower uncertainties than $10^{-4}\,\mathrm{rad}$. Hence, regular in-flight updating of these parameters may not be recommended, as those updates may introduce unnecessary jitter without any benefit to the overall accuracy of the magnetometer calibration.

## 7 Further Discussion and Conclusions

The orthogonalization angles are known to be relatively stable when compared to the spin axis direction angles. Fortunately, as shown in the previous section, the spin axis angles can be updated with high accuracy more regularly than the sensor elevation angles $\delta\theta_{\mathrm{S1}}$ and $\delta\theta_{\mathrm{S2}}$. The parameter decoupling introduced in section 2 pays off here, as spin axis variations do not require re-determination of the sensor elevation angles as would be the case when using the calibration Equation (1) with the coupling matrix (2) instead of Equation (9).

It should be noted that both Equations (1) and (9) assume raw magnetometer outputs to be linearly transformable into accurate magnetic field estimates. This assumption of linearity can only be fulfilled to a certain degree when dealing with actual magnetometer hardware. Non-linearities (e.g., Auster et al., 2008; Russell et al., 2016) will adversely affect the calibration as described here if not characterized, quantified, and corrected beforehand, as they produce spin tone and higher harmonic signals in the magnetic field measurements. THEMIS FGM data, for instance, suffer from slight non-linearities in digital-to-analogue converters that are part of the magnetometer hardware. These are known from ground characterization of the instruments and are routinely corrected in advance of any in-flight calibration activities and/or any conversion of magnetometer outputs into calibrated magnetic field measurements (Auster et al., 2008).

Assuming instrument linearity, the uncertainty-based approach to determining the spin-related calibration parameters allows for a meaningful estimation of the error alongside with any parameter updates. These errors can be compared to the uncertainties of the already known parameters, determined either on ground or in-flight. Therewith, it is possible to decide whether any update of the calibration parameters is necessary/advised or, instead, would just introduce unnecessary variations in the calibration parameters over time.





In addition, the availability of calibration parameter estimates associated to low uncertainties, sufficient in number and quality, determines what is possible in terms of cadence of parameter updates. This availability depends on the orbit of the spacecraft (the presence in regions of certain field conditions), and also on the spin period of the spacecraft. In general, short spin periods (high spacecraft spin frequencies) are favorable, as they increase the number of spins that may be taken into

account in subintervals of certain length. A larger number of spin periods reduces the influence of natural field fluctuations at (double) the spin frequency, while short subinterval lengths ensure the constancy of the parameters and environmental conditions. In the given THEMIS-C example, the spin plane offsets $O_{S1/2}$ may be continuously tracked while the spacecraft remained in the solar wind and in the low field parts of the magnetosphere. The spin axis components $\sigma_{Px/y}$, the gain ratio $g$ and azimuthal orthogonalization angle $\phi_{S12}$ can easily be determined separately before/after each perigee pass, whereas

accurate determinations of the elevation angles $\theta_{S1/2}$ may only be possible when taking into account estimates from several spacecraft orbits.

Finally we would like to note that the benefits of parameter decoupling (i.e., a sensible choice of parameters when taking into account the behavior of the magnetometer and spacecraft hardware) and of the uncertainties-based determination of those parameters are not tied to the exact definitions of the calibration Equation (9) and matrices (4) to (7). For example, the offsets

may be applied in an orthogonal, spacecraft-fixed coordinate system instead of in the sensor coordinate system, if the main contribution to the offsets is expected from spacecraft stray fields at the sensor position. The order of the gain, orthogonalization, and alignment matrices (here $\mathbf{G}$, $\mathbf{\Gamma}$, $\mathbf{\Sigma}$, and $\mathbf{\Phi}$) may be changed, and/or the 12 degrees of freedom of the calibration parameters may be distributed over a larger number of matrices and offset vectors to account for changes pertaining to different parts of the magnetometer/spacecraft system in different coordinate systems (e.g., see equations in Fornaçon et al., 1999; Auster et al.,

2008). Hence, while following the principles set out in this paper, a different set of calibration parameters and corresponding calibration equation may be specifically selected for each magnetometer/spacecraft combination.

*Data availability.* Data from the THEMIS mission including FGM and ESA data are publicly available from the University of California Berkeley and can be obtained from http://themis.ssl.berkeley.edu/data/themis.

*Competing interests.* The authors declare that they have no conflict of interest.

*Acknowledgements.* We acknowledge NASA contract NAS5-02099 and Vassilis Angelopoulos for use of data from the THEMIS mission. Specifically, we acknowledge Charles W. Carlson and James P. McFadden for use of ESA data and Karl-Heinz Glassmeier, Hans-Ulrich Auster and Wolfgang Baumjohann for the use of FGM data provided under the lead of the Technical University of Braunschweig and with financial support through the German Ministry for Economy and Technology and the German Center for Aviation and Space (DLR) under contract 50 OC 0302.



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
