# Peer review of "Advanced calibration of magnetometers on spin-stabilized spacecraft based on parameter decoupling"

_Geoscientific Instrumentation, Methods and Data Systems, 2018_

## Referee Comment (RC1) · Anonymous Referee #1 · 3 Oct 2018

Comments on "Advanced calibration of magnetometers on spin-stabilized spacecraft based on parameter decoupling" by Ferdinand Plaschke, Hans-Ulrich Auster, David Fischer, Karl-Heinz Fornaçon, Werner Magnes, Ingo Richter, Dragos Constantinescu, and Yasuhito Narita (Geoscientific Instrumentation, Methods and Data Systems (GI), gi-2018-45)

The paper concerns an additional aspect of FGM application onboard the rotating spacecrafts and is useful for magnetic field measurements in space plasma. Major remarks: For such a serious problem as space magnetic field measurements the MS contains non-metrological definitions as "approximately", "slightly over" etc. at estima-

tion of physical values and approximation procedures. See, for example: p. 4, lines 15-16: "... the spin axis is assumed to be approximately aligned with the Pz = S3 axis" [The admissible errors should be estimated] ; p. 6, lines 10-15, Eqs. (11)-(15). [The admissible approximation errors should be estimated]; p. 7, line 11: "... and further dropping second order factors, "[The approximation errors should be indicated]; p. 11, line 1: "... slightly over and under the spin frequency: $\omega\pm$ = $2\pi n\pm$/tint with n$\pm$ âĹĹ N and slightly over/under n ..." [The admissible intervals should be estimated]; p. 11, line 20: "... slightly over/under n." [The admissible intervals should be estimated]; p. 11, line 28: "... can be omitted due to linearization." [The approximation errors should be indicated] ; p. 11, line 32: "... if the initial set of calibration parameters is not too inaccurate." [The admissible approximation errors should be indicated] ; [During linearization procedure (see, for example, p.6, lines 10-15; p.11, line 28) the basic equations are simplified what leads to the appearance of additional errors at data processing. So, the errors of such an approximation should be estimated, at least for main cases]. P. 11, lines 6-9. It is unclear, how to de-trend the B data, i. e. to separate the studied process and linear trend with given error. P. 13, line 8. What does it mean "... as temperatures relax to stationary values." in practical sense, i. e. admissible unbalance between stationary value and real unsteady temperature after eclipse, for example? What level was assumed by authors during data processing? It should be clearly indicated. P. 10, line 27. Why for Bp "...the minimal modulus of the spin plane field over the subinterval: $\min(\sqrt{(B\_x^2 + B\_y^2)})$." was chosen? It seems to be the better value is $\mathrm{avg}(\sqrt{(B\_x^2 + B\_y^2)})$. Numerical example: Gp = 0.999 Ga = 1.0001 g = 1.002 thetaS1 = pi/2+0.001 thetaS2 = pi/2-0.0015 phiS12 = pi/2+0.002 sigmaPx = 0.0008 sigmaPy = -0.0012

Normalized magnetic field in the non-spinning (inertial), orthogonal, spin-axis aligned (Z = z) coordinate system BX = 0.3 BY = 0.8 BZ $\approx$ -0.5196

Spin frequency is 2*pi The test data was generated for 5 rotation periods with the time discretization dt = 0.2 True value Bp = $\sqrt{(B\_x^2 + B\_y^2)}$ = 0,85440 The estimation of Bp using the minimum of the modulus of the B' projection on plane XOY (the rotation

plane) gives the value dBp_min = Bp − min($\sqrt{(B\_x\char`\^'2\ (t) + B\_y\char`\^'2\ (t))}$) = 0,00236270
The estimation of Bp at use of the average value of the modulus for the B' projection on
plane XOY gives the value dBp_avg = Bp − avg($\sqrt{(B\_x\char`\^'2\ (t) + B\_y\char`\^'2\ (t))}$) = 0,00087464
Finally (dB_p_min)/(dB_p_avg)=2.7013

Minor remarks: P.5, line 16, Eq. (6): No definition of angle $\varphi$a P.9, Table 2, group 1:
No definition of Fa P.9, line 20, p.10, line 26: No definition of Fa P.9, Table 2, group 3:
it should be OS1 and OS2 instead of OS1 and OS1 P.10, line 12: it should be GT(Ts,
Te) instead of GT(Ts, Ts) P.13, line 16: it should be FGM instead of FGL.

---

## Referee Comment (RC2) · Anonymous Referee #2 · 12 Dec 2018

To begin, this is an excellent paper.

The paper describes an in situ calibration methodology for spacecraft vector magnetometers, which this reader views as a process of four steps from ambient field to despun discrete time series data. These four steps are:

1. signal collection by a spinning magnetometer system, which is a full quadrature amplitude modulation.

2. disturbances to the signal in the form of offsets, gain and alignment errors

3. inverse transformation to correct for disturbances

4. despinning of the data, which is a full quadrature amplitude demodulation.

The paper is largely about determining the parameters for step 3, the inverse transformation. The determination process is that of a typical inverse problem where the the minor terms of eq's 24-26 serve as the residuals. Iterative processing is normally used to minimize the residuals, which the authors indicate as having occurred here. The process has been further optimized by the use of unbiased selection of data segments likely to yield best inverse transformation parameter estimates.

Any mismatch between the disturbances and the inverse transformation leads to data errors in the form of carrier feed-through and non-zero frequency sum and difference components which manifest as baseband and 2nd harmonic terms. These terms are highlighted in equations 24-26, and in Section 4.

Equations (20,21)

Although it is entirely obvious that $\omega t$ is the spin parameter, to be consistent with the excellent presentation of parameter definitions up to this point, $\omega t$ could beneficially be defined here. As the paper is written this spin parameter appears to apply to both steps 1 and 4, the amplitude modulations and demodulations. In the case that the modulation frequencies are identical the situation can be called synchronous demodulation, which has huge advantages. In general though, the modulation and demodulation frequencies are not identical, but are determined and matched very accurately. As the spacecraft ages and fuel is used, not only are spin angles changing, but so are moments of inertia. The spin frequency cannot be determined from the short analysis periods, due to well understood uncertainty principles. As noted earlier in the paper the spin frequency/angle were determined as a priors, using the IGRF and other tools. Some brief discussion is recommended as to the definition of $\omega$ and $\omega t$, and how they relate to the amplitude demodulation parameters. Also, for the determination of g, the X/Y gain ratio, the choices of +/-15% [page 15] of the carrier/spin frequency seem large in context of equations 30 and 31. Is this a requirement of the data rate?

Equations (24,25)

Each of these equations consists of five terms, pair-wise similar. As this is the end of section 3, for the general reader some brief summary descriptions could be useful. The first terms are the primary measurement terms, while the second and third terms represent (roughly) the modulations of the X/Y offsets and the projection from the spin axis into the spin tone. The fourth and fifth terms are the 2nd harmonic terms resulting mainly from the differential gains of the X/Y channels. A quick review of the physical mechanism leading to a detectable 2nd harmonic term would be useful as the reader at this point needs a clear understanding that he or she is considering only despun data. Section 4 goes into all of this in further detail.

The use of rotations in fields as the basis for a calibration methodology now has a long history. This reader's first contact with such ideas was that of A.W. Green Jr, in the 1980's. Green took on the much easier task of calibrating observatory magnetometers, for which multiple rotation axes were possible. Green's description of his method can be found as doi.org/10.1016/0031-9201(90)90217-L . Such methods have come a long way.
* * *

---

## Author Comment (AC1) · 30 Jan 2019

**Response to reviewers' comments**

First of all, we would like to thank the reviewers for their helpful comments. In this document, the reviewers' comments are given in boldface and our answers are in normal type. Line and section numbers refer to the original manuscript.

**Reviewer #1**

Major remarks:

Comment 1: For such a serious problem as space magnetic field measurements the MS contains nonmetrological definitions as "approximately", "slightly over" etc. at estimation of physical values and approximation procedures. See, for example:

- 1. p. 4, lines 15-16: "... the spin axis is assumed to be approximately aligned with the Pz = S3 axis" [The admissible errors should be estimated.]
- 2. p. 6, lines 10-15, Eqs. (11)-(15). [The admissible approximation errors should be estimated.]
- **3.** p. 7, line 11: "... and further dropping second order factors, " [The approximation errors should be indicated.]
- 4. p. 11, line 1: "... slightly over and under the spin frequency: ... with ... and slightly over/under n ..." [The admissible intervals should be estimated.]
- 5. p. 11, line 20: "... slightly over/under n." [The admissible intervals should be estimated.]
- 6. p. 11, line 28: "... can be omitted due to linearization." [The approximation errors should be indicated.]
- 7. p. 11, line 32: "... if the initial set of calibration parameters is not too inaccurate." [The admissible approximation errors should be indicated.]
- 8. [During linearization procedure (see, for example, p.6, lines 10-15; p.11, line 28) the basic equations are simplified what leads to the appearance of additional errors at data processing. So, the errors of such an approximation should be estimated, at least for main cases.]

Points 1, 2, 3, 6, 7, and 8 deal with errors associated to the linearization procedure and/or in the corresponding assumptions regarding the calibration parameters. The aim of the linearization procedure is to obtain the uncertainties  $\Delta\sigma_{Px/y}$ ,  $\Delta g$ ,  $\Delta\phi_{S12}$ ,  $\Delta O_{S1/2}$ , and  $\Delta\theta_{S1/2}$  in calibration parameter determination under different conditions, shown in the right column of Table 2. The actual calibration parameter estimates, however, are unaffected by these errors, as they are obtained by using the full – not linearized – calibration Equation (9), with matrices (4) to (7). The question is, hence, how much the parameters g,  $G_p$ ,  $G_a$ ,  $\sigma_{Px}$ ,  $\sigma_{Py}$ ,  $\delta\theta_{S1}$ ,  $\delta\phi_{S12}$  may deviate from the assumed values (see Equations 11 to 15) until the errors in the uncertainties  $\Delta\sigma_{Px/y}$ ,  $\Delta g$ ,  $\Delta\phi_{S12}$ ,  $\Delta O_{S1/2}$ , and  $\Delta\theta_{S1/2}$  in Table 2 become unacceptably large.

As shown in the application section 6, we are only interested in the orders of magnitude of the uncertainties (a factor of 10). Conservatively limiting the error to a factor of 2, and taking into account that relative errors from individual matrices (16) to (19) may increase by multiplication due to Equation (10), we obtain individual limits on the relative deviations D of the parameters of:  $2 = D^4$ , D = 1.189, i.e., 19%.

Such an error is extraordinarily large when compared to the accuracy in the knowledge of the calibration parameters and of the geometry of the spacecraft, even before performing any in-flight calibration. For the angles  $\sigma_{Px}$ ,  $\sigma_{Py}$ ,  $\delta\theta_{s1}$ ,  $\delta\theta_{s1}$ , and  $\delta\varphi_{s12}$ , this means deviations from 0 by up to 11° are

acceptable (arcsin(19%)). Note that alignment uncertainties should usually be lower than 1°. For the gains g,  $G_p$ , and  $G_a$ , errors of 19% would be theoretically acceptable. Ground calibration, however, should reduce these errors to less than 0.1%. Hence, the assumptions made due to linearization (e.g., Equations 11 to 15) are not restrictive at all, and easily fulfilled in practice.

Points 4 and 5 deal with estimating amplitudes of natural magnetic field fluctuations in the vicinity of the spin frequency or second harmonic. Slightly over/under n transforms into frequencies that are "slightly" over or under the first/second harmonic frequency. The question is how large "slightly" needs to be. It should be large enough so that an increased amplitude at the spin frequency or second harmonic does not leak significantly into the selected frequency bins. It should also be small enough so that the amplitudes determined at the selected frequencies resemble the natural amplitude level at the spin frequency or second harmonic.

The ratio of spin tone leakage with respect to natural fluctuations around the spin frequency is a function of spin tone amplitude and the Fourier transform window. Choosing the optimal frequencies above and below omega would require a recalculation for each subinterval considered. In practice, experience from THEMIS and MMS calibration activities shows that frequencies 15% of omega above and below omega and 2\*omega are a good choice. The reason is that ±15% of omega is a large "distance" in frequency with respect to the sharp spin tone signatures from incorrect calibration parameter choice and leakage, yet 15%\*omega is still a small frequency interval for natural fluctuations. These relative frequencies have also been used in the THEMIS example presented in section 6.

We opted for including a general discussion on admissible errors between sections 6 and 7 (new section 7). A discussion on choices in frequency above/below omega is added in page 11, line 5, after Equations (30) and (31).

**Comment 2: P. 11, lines 6-9. It is unclear, how to de-trend the B data, i. e. to separate the studied process and linear trend with given error.**

The Fourier transform is applied either to the spin axis component data or to the modulus of the spin plane field. That component/modulus is detrended by simply subtracting a linear fit to it, as we now state in the paper.

**Comment 3: P. 13, line 8. What does it mean "... as temperatures relax to stationary values." in practical sense, i. e. admissible unbalance between stationary value and real unsteady temperature after eclipse, for example? What level was assumed by authors during data processing? It should be clearly indicated.**

The admissible unbalance is very spacecraft and instrument specific. Some instruments/spacecraft show significant changes in magnetometer calibration parameters with just a few degrees change in sensor/electronics/spacecraft temperatures. At other spacecraft, almost no changes are apparent, although temperatures vary by several tens of degrees. During processing of the THEMIS spacecraft data we excluded eclipse and high field magnetometer saturation intervals. During the remaining time intervals, the electronics and sensor temperatures vary within 3 degrees, which in the THEMIS case is totally insignificant for the calibration parameters. We state this now at the end of page 14.

Comment 4: P. 10, line 27. Why for Bp "... the minimal modulus of the spin plane field over the subinterval:  $min(p(B_x^2 + B_y^2))$ ." was chosen? It seems to be the better value is  $avg(p(B_x^2 + B_y^2))$ .

Numerical example: Gp = 0.999, Ga = 1.0001, g = 1.002, thetaS1 = pi/2+0.001, thetaS2 = pi/2-0.0015, phiS12 = pi/2+0.002, sigmaPx = 0.0008, sigmaPy = -0.0012. Normalized magnetic field in the non-spinning (inertial), orthogonal, spin-axis aligned (Z = z) coordinate system: BX = 0.3, BY = 0.8, BZ = -0.5196. Spin frequency is 2\*pi. The test data was generated for 5 rotation periods with the time discretization dt = 0.2. True value Bp =  $p(B_x^2 + B_y^2 2) = 0.85440$ . The estimation of Bp using the minimum of the modulus of the B' projection on plane XOY (the rotation plane) gives the value dBp\_min = Bp - min( $p(B_x^2 (t) + B_y^2 (t))) = 0.00236270$ . The estimation of Bp at use of the average value of the modulus for the B' projection on plane XOY gives the value dBp\_avg = Bp -  $avg(p(B_x^2 (t) + B_y^2 (t))) = 0.00087464$ . Finally (dB\_p\_min)/(dB\_p\_avg)=2.7013.

We agree that the slightly better value is  $avg(p(B_x^2 + B_y^2))$ . However, dBp\_avg and dBp\_min are both very small in comparison to Bp, which is the quantity we are interested in here, and we considered  $min(p(B_x^2 + B_y^2))$  to be the more conservative choice, as stated in the paper.

**Minor remarks:**

**Comment 6: P.5, line 16, Eq. (6): No definition of angle 'a.**

As stated in the sentence above the equation, phi\_a is a rotation angle about the spin axis.

**Comment 7: P.9, Table 2, group 1: No definition of Fa.**

**Comment 8: P.9, line 20, p.10, line 26: No definition of Fa.**

The definition is stated on page 9 line 19. F\_a is then exactly defined in Equation (31).

**Comment 9: P.9, Table 2, group 3: it should be OS1 and OS2 instead of OS1 and OS1.**

**Comment 10: P.10, line 12: it should be GT(Ts, Te) instead of GT(Ts, Ts).**

Thank you very much for pointing out these typos. They are now corrected.

**Comment 11: P.13, line 16: it should be FGM instead of FGL.**

In this case, FGL is correct. It is the name of the data product and not the acronym for Fluxgate Magnetometer, the instrument.

**Reviewer #2**

**Comment 12: Equations (20,21)**

Although it is entirely obvious that (omega t) is the spin parameter, to be consistent with the excellent presentation of parameter definitions up to this point, (omega t) could beneficially be defined here. As the paper is written this spin parameter appears to apply to both steps 1 and 4, the

amplitude modulations and demodulations. In the case that the modulation frequencies are identical the situation can be called synchronous demodulation, which has huge advantages. In general though, the modulation and demodulation frequencies are not identical, but are determined and matched very accurately. As the spacecraft ages and fuel is used, not only are spin angles changing, but so are moments of inertia. The spin frequency cannot be determined from the short analysis periods, due to well understood uncertainty principles. As noted earlier in the paper the spin frequency/angle were determined as a priors, using the IGRF and other tools. Some brief discussion is recommended as to the definition of omega and (omega t), and how they relate to the amplitude demodulation parameters.

Thank you for the suggestion. We now define omega t right after Equations (20) to (22).

The spin rate is not usually determined from magnetic field measurements, but from independent sun sensor or star tracker measurements. Hence, the spin rate is adjusted regularly and is known with very high accuracy. Exact knowledge of this frequency is highly important for spin demodulation, which is done in a later part of instrument data processing. However, the estimate of the calibration parameters does not rely on exact demodulation, but only on the amplitude of certain frequencies within the Fourier transformed signal. Small shifts below the frequency resolution of the DFT will therefore only result in a slightly smaller sensitivity, as the amplitude of the considered frequency is reduced by a small amount.

**Comment 13: Also, for the determination of g, the X/Y gain ratio, the choices of +/-15% [page 15] of the carrier/spin frequency seem large in context of equations 30 and 31. Is this a requirement of the data rate?**

See answer to comment 1, points 4 and 5: The large separations between frequencies are there to avoid significant leakage from the signals at omega and 2\*omega.

**Comment 14: Equations (24,25)**

Each of these equations consists of five terms, pair-wise similar. As this is the end of section 3, for the general reader some brief summary descriptions could be useful. The first terms are the primary measurement terms, while the second and third terms represent (roughly) the modulations of the X/Y offsets and the projection from the spin axis into the spin tone. The fourth and fifth terms are the 2nd harmonic terms resulting mainly from the differential gains of the X/Y channels. A quick review of the physical mechanism leading to a detectable 2nd harmonic term would be useful as the reader at this point needs a clear understanding that he or she is considering only despun data. Section 4 goes into all of this in further detail.

As suggested, we have added a discussion on Equations (24) to (26) at the end of section 3.

Comment 15: The use of rotations in fields as the basis for a calibration methodology now has a long history. This reader's first contact with such ideas was that of A.W. Green Jr, in the 1980's. Green took on the much easier task of calibrating observatory magnetometers, for which multiple rotation axes were possible. Green's description of his method can be found as doi.org/10.1016/0031-9201(90)90217-L. Such methods have come a long way.

Thank you very much for making us aware of this reference. We cite it now in the introduction.

**Advanced calibration of magnetometers on spin-stabilized spacecraft based on parameter decoupling**

Ferdinand Plaschke1, Hans-Ulrich Auster2, David Fischer1, Karl-Heinz Fornaçon2, Werner Magnes1, Ingo Richter2, Dragos Constantinescu2, and Yasuhito Narita1

1Space Research Institute, Austrian Academy of Sciences, Graz, Austria.

2Institute for Geophysics and Extraterrestrial Physics, Braunschweig University of Technology, Braunschweig, Germany.

Correspondence: Ferdinand Plaschke (ferdinand.plaschke@oeaw.ac.at)

**Abstract.** Magnetometers are key instruments onboard spacecraft that probe the plasma environments of planets and other solar system bodies. The linear conversion of raw magnetometer outputs to fully calibrated magnetic field measurements requires the accurate knowledge of 12 calibration parameters: 6 angles, 3 gain factors, and 3 offset values. The in-flight determination of 8 of those 12 parameters is enormously supported if the spacecraft is spin stabilized, as an incorrect choice of those parameters

- 5 will lead to systematic spin harmonic disturbances in the calibrated data. We show that published equations and algorithms for the determination of the 8 spin-related parameters are far from optimal, as they do not take into account the physical behavior of science-grade magnetometers and the influence of a varying spacecraft attitude on the in-flight calibration process. Here, we address these issues. Based on decades-long developments and experience in calibration activities at the Braunschweig University of Technology, we introduce advanced calibration equations, parameters, and algorithms. With their help, it is
- 10 possible to decouple different effects on the calibration parameters, originating from the spacecraft or the magnetometer itself. A key point of the algorithms is the bulk determination of parameters and associated uncertainties. Lowest uncertainties are expected under parameter specific conditions. By application to THEMIS-C magnetometer measurements, we show where these conditions are fulfilled along a highly elliptical orbit around Earth.

**1 Introduction**

The investigation of the plasma environment in the heliosphere, around planets, moons, comets, or other solar system bodies, requires accurate in-situ observations of the magnetic field. Magnetometers on board spacecraft can provide these key measurements if accurately calibrated on ground and in flight. The calibration process delivers the parameters needed to convert raw magnetometer measurements into magnetic field observations *B* = (*Bx*, *By*, *Bz*)T in physically meaningful coordinate systems and units (usually nanotesla: nT). Commonly, a linear calibration equation is applied for this conversion (e.g., Fornaçon et al., 1999; Balogh et al., 2001b; Auster et al., 2008):

$$\boldsymbol{B} = \mathbf{C} \cdot (\boldsymbol{B}_{\mathrm{S}} - \boldsymbol{O}_{\mathrm{S}}) \tag{1}$$

Here  $B_{\rm S} = (B_{\rm S1}, B_{\rm S2}, B_{\rm S3})^{\rm T}$  is the raw magnetometer output in non-orthogonal sensor coordinates,  $O_{\rm S}$  corrects for non-vanishing magnetometers outputs in zero ambient fields (so-called offsets, which include spacecraft-generated magnetic fields

at the sensor position), and C is the  $3 \times 3$  coupling matrix. This matrix may have the following form (e.g., Kepko et al., 1996):

$$\mathbf{C} = \begin{pmatrix} \sin\theta_1 \cos\phi_1 & \sin\theta_1 \sin\phi_1 & \cos\theta_1 \\ \sin\theta_2 \cos\phi_2 & \sin\theta_2 \sin\phi_2 & \cos\theta_2 \\ \sin\theta_3 \cos\phi_3 & \sin\theta_3 \sin\phi_3 & \cos\theta_3 \end{pmatrix}^{-1} \cdot \begin{pmatrix} G_{\mathrm{S}1} & 0 & 0 \\ 0 & G_{\mathrm{S}2} & 0 \\ 0 & 0 & G_{\mathrm{S}3} \end{pmatrix}$$
(2)

The coupling matrix C depends on three scaling factors (GS1, GS2, and GS3, also called the gains) and six angles (θ1, θ2, θ3, and φ1, φ2, φ3) which define the directions of the three sensor axes in the orthogonal coordinate system to which B pertains.
Calibrating a magnetometer means finding the three gains, six angles, and three offset components (i.e., in total 12 parameters)

so that  $\boldsymbol{B}$  can accurately be obtained from  $\boldsymbol{B}_{\mathrm{S}}$ .

Ground calibration of magnetometers is facilitated by rotating them in Earth's magnetic field (Green, 1990). Similarly, operating a 
[revised manuscript text omitted]

$$\boldsymbol{\Gamma} = \begin{pmatrix} \sin\theta_{\mathrm{S}1} & 0 & \cos\theta_{\mathrm{S}1} \\ \cos\phi_{\mathrm{S}12}\sin\theta_{\mathrm{S}2} & \sin\phi_{\mathrm{S}12}\sin\theta_{\mathrm{S}2} & \cos\theta_{\mathrm{S}2} \\ 0 & 0 & 1 \end{pmatrix}^{-1}$$
(4)

Here,  $\theta_{S1}$  and  $\theta_{S2}$  are the angles between the sensor axes S1 and S2 with respect to S3=Pz, and  $\phi_{S12}$  is the angle between the projections of S1 and S2 onto a plane perpendicular to S3, the Px-Py plane. Note that S1 lies in the Px-Pz plane (see Figure 1a).

In the next step, the orientation of that sensor package system needs to be defined in a spacecraft-fixed spin axis aligned coordinate system. This latter transformation is expected to change every time there is a maneuver of the spacecraft, as fuel consumption will change the tensor of inertia and, thus, the spin axis direction in any spacecraft fixed coordinate system. The spin axis direction can be defined in the orthogonal sensor package system using two parameters or angles. During maneuvers,

10

5

only those two parameters/angles should change, because the geometry inside the sensor package should not be affected. A rotation matrix  $\Sigma$  into a spin axis aligned coordinate system dependent on the two angles  $\sigma_{Px}$  and  $\sigma_{Py}$  can be defined as follows:

$$\boldsymbol{\Sigma} = \begin{pmatrix} \cos\sigma_{\mathrm{P}x} & 0 & -\sin\sigma_{\mathrm{P}x} \\ 0 & 1 & 0 \\ \sin\sigma_{\mathrm{P}x} & 0 & \cos\sigma_{\mathrm{P}x} \end{pmatrix} \cdot \begin{pmatrix} 1 & 0 & 0 \\ 0 & \cos\sigma_{\mathrm{P}y} & -\sin\sigma_{\mathrm{P}y} \\ 0 & \sin\sigma_{\mathrm{P}y} & \cos\sigma_{\mathrm{P}y} \end{pmatrix}$$
(5)

15

Here,  $\sigma_{Py}$  is the angle between Pz and the projection of the spin axis onto the Py-Pz-plane, positive towards Py;  $\sigma_{Px}$  is the angle between that projection and the spin axis, positive towards Px. The angles are illustrated in Figure 1b. Note that the spin axis is assumed to be approximately aligned with the Pz = S3 axis. As a result, the angles  $\sigma_{Px}$  and  $\sigma_{Py}$  will be small and can be associated with the Px and Py coordinates of a unit vector that points in spin axis direction.

Using the angles  $\sigma_{Px}$  and  $\sigma_{Py}$  to define the spin axis direction is advantageous over using the angles  $\theta_3$  and  $\phi_3$ , as the latter angle is badly defined if  $\theta_3$  is small. Furthermore, it should also be noted that a change in direction of the spin axis requires an update of all angles of matrix  $\Theta$  as defined above, even though the magnetometer (sensor) itself is unaffected. Only two parameters ( $\sigma_{Px}$  and  $\sigma_{Py}$ ) need to be changed here to adapt the matrix  $\Sigma$  to the new spin axis direction.

5

To completely orient the sensor package (system) in the spin axis aligned coordinate system, a rotation about the spin axis (rotation matrix  $\Phi$ ) also needs to be taken into account:

$$\boldsymbol{\Phi} = \begin{pmatrix} \cos\phi_{\mathbf{a}} & -\sin\phi_{\mathbf{a}} & 0\\ \sin\phi_{\mathbf{a}} & \cos\phi_{\mathbf{a}} & 0\\ 0 & 0 & 1 \end{pmatrix}$$
(6)

As we will show later, this rotation does not affect the spin tone content in the despun magnetic field observations. The angle is affected by the orientation of a magnetometer boom and may change due to boom bending (Farrell et al., 1995).

10

15

Altogether, we can replace the orthogonalization and reorientation matrix  $\Theta$  by  $\Phi \cdot \Sigma \cdot \Gamma$  in Equation (3). Let's focus then again on the gain matrix **G** in that equation. As mentioned in the introduction, the spacecraft spin aids the determination of the ratio  $g^2 = G_{S1}/G_{S2}$  of the spin plane gains, but not the absolute gains in the spin plane  $G_p = \sqrt{G_{S1}G_{S2}}$  and along the spin axis  $G_a = G_{S3}$ . Hence, it makes sense to use the parameters g and  $G_p$  instead of  $G_{S1}$  and  $G_{S2}$  in the matrix **G**, to decouple parameters that can be frequently updated from parameters that are only obtainable in flight from comparison to model fields or measurements of other instruments:

$$\mathbf{G} = \begin{pmatrix} gG_{\rm p} & 0 & 0\\ 0 & G_{\rm p}/g & 0\\ 0 & 0 & G_{\rm a} \end{pmatrix}$$
(7)

Note that Kepko et al. (1996) use the difference of the inverse gains  $\Delta G_{21} = 1/G_{S2} - 1/G_{S1}$  instead of g. However, later changes in the absolute gains  $G_{S1}$  and  $G_{S2}$  then require necessarily an update of  $\Delta G_{21}$  in order to avoid perturbations at the second harmonic in the despun data. The gain ratio g, instead, is decoupled from changes in the absolute gains  $G_p$  and  $G_a$ .

20

The gains should be stable parameters in the absence of temperature variations. These variations in the gains can be determined from ground calibration, resulting in a diagonal gain correction matrix  $\mathbf{G}_{\mathrm{T}}(T_{\mathrm{s}}, T_{\mathrm{e}})$  that is dependent on the magnetometer sensor  $(T_{\mathrm{s}})$  and electronics  $(T_{\mathrm{e}})$  temperatures. That matrix should be directly applied to the magnetometer output  $\boldsymbol{B}_{\mathrm{S}}$ , requiring the knowledge of the sensor and electronics temperatures:

$$\boldsymbol{B}_{\mathrm{ST}} = \boldsymbol{G}_{\mathrm{T}}(T_{\mathrm{s}}, T_{\mathrm{e}}) \cdot \boldsymbol{B}_{\mathrm{S}}$$

$$\tag{8}$$

25 The resulting temperature corrected output  $B_{ST}$  may then be further converted to B via the coupling matrix  $\mathbf{C} = \mathbf{\Phi} \cdot \boldsymbol{\Sigma} \cdot \boldsymbol{\Gamma} \cdot \mathbf{G}$ and the offset vector O using Equation (1), after replacing  $B_S$  with  $B_{ST}$ . This also has the advantage that the further applied absolute gains  $G_p$  and  $G_a$  and the gain ratio  $g^2$  should all be approximately 1 and unitless.

Altogether, we suggest to use the following improved calibration equation:

$$\boldsymbol{B} = \boldsymbol{\Phi} \cdot \boldsymbol{\Sigma} \cdot \boldsymbol{\Gamma} \cdot \boldsymbol{G} \cdot (\underbrace{\boldsymbol{G}_{\mathrm{T}}(T_{\mathrm{s}}, T_{\mathrm{e}}) \cdot \boldsymbol{B}_{\mathrm{S}}}_{=\boldsymbol{B}_{\mathrm{ST}}} - \boldsymbol{O}_{\mathrm{S}})$$
(9)

with matrices defined in Equations (4) to (7) instead of the simpler Equations (1) and (2). The parameters whose determination is supported by the spacecraft spin are:  $\theta_{S1}$ ,  $\theta_{S2}$ ,  $\phi_{S12}$ ,  $\sigma_{Px}$ ,  $\sigma_{Py}$ , g,  $O_{S1}$ , and  $O_{S2}$ .

**3 Calibration Parameter Influence on Spin Tone Harmonics**

15

To determine the influence of the calibration parameters on the spin tone harmonic disturbances in the despun magnetic field 5 measurements, we use a similar mathematical approach to Kepko et al. (1996) in this section. Based on the results, we go on to derive the optimal conditions for the determination of each parameter in section 4.

First, we compute the temperature corrected sensor output  $B_{ST}$  as a function of the external field B in the spinning coordinate system:

$$\boldsymbol{B}_{\mathrm{ST}} = \mathbf{G}^{-1} \cdot \boldsymbol{\Gamma}^{-1} \cdot \boldsymbol{\Sigma}^{-1} \cdot \boldsymbol{\Phi}^{-1} \cdot \boldsymbol{B} + \boldsymbol{O}_{\mathrm{S}}$$
(10)

10 We linearize all the matrices, using the following simplifying assumptions. The validity of these assumptions and the admissible deviations are discussed in section 7.

$$g \approx 1, \qquad G_{\rm p} \approx 1, \qquad G_{\rm a} \approx 1$$
 (11)

$$\sigma_{\mathrm{P}x} \approx 0, \qquad \qquad \sigma_{\mathrm{P}y} \approx 0 \tag{12}$$

$$\theta_{S1} \approx \pi/2, \quad \delta \theta_{S1} = \theta_{S1} - \pi/2 \approx 0$$
(13)

$$\theta_{S2} \approx \pi/2, \quad \delta \theta_{S2} = \theta_{S2} - \pi/2 \approx 0$$
(14)

$$\phi_{S12} \approx \pi/2, \quad \delta \phi_{S12} = \phi_{S12} - \pi/2 \approx 0$$
(15)

Furthermore, we assume  $\phi_a \approx 0$  without loss of generality. Dropping second order factors, we obtain the following linearized inverted matrices used in Equation (10):

$$\mathbf{G}^{-1} = \begin{pmatrix} 1/(gG_{\rm p}) & 0 & 0\\ 0 & g/G_{\rm p} & 0\\ 0 & 0 & 1/G_{\rm a} \end{pmatrix}$$
(16)

$$\quad \Gamma^{-1} = \begin{pmatrix} 1 & 0 & -\delta\theta_{S1} \\ -\delta\phi_{S12} & 1 & -\delta\theta_{S2} \\ 0 & 0 & 1 \end{pmatrix}$$
(17)

$$\Sigma^{-1} = \begin{pmatrix} 1 & 0 & \sigma_{\mathrm{P}x} \\ 0 & 1 & \sigma_{\mathrm{P}y} \\ -\sigma_{\mathrm{P}x} & -\sigma_{\mathrm{P}y} & 1 \end{pmatrix}$$
(18)

$$\Phi^{-1} = \begin{pmatrix} 1 & \phi_{a} & 0 \\ -\phi_{a} & 1 & 0 \\ 0 & 0 & 1 \end{pmatrix}$$
(19)

Furthermore, without loss of generality, we assume the magnetic field in the despun (inertial) coordinate system (directions X, Y, and Z) to be in the X-Z-plane, and the spacecraft to spin around the Z-axis, which corresponds with the z-axis in the spacecraft fixed, spin aligned coordinate system (see Table 1). In that latter system, the field rotates and has the following form:

$$B_x = B_p \cos \omega t = B_X \cos \omega t \tag{20}$$

$$\quad B_y = B_p \sin \omega t = B_X \sin \omega t \tag{21}$$

$$B_z = B_a = B_Z \tag{22}$$

Here,  $\omega$  is the angular frequency of the spacecraft rotation, usually determined from sun sensor or star tracker measurements, and t denotes the time. Inserting these relations in Equation (10) yields the expected temperature corrected output of the magnetometer in sensor coordinates. By applying the despin rotation matrix

$$\quad \mathbf{D} = \begin{pmatrix} \cos \omega t & -\sin \omega t & 0\\ \sin \omega t & \cos \omega t & 0\\ 0 & 0 & 1 \end{pmatrix}$$
(23)

to Equation (10) to transform  $B_{ST}$  into a non-orthogonal, despun coordinate system (directions X', Y', and Z', see Table 1), after sorting by frequency and phase of the terms, and further dropping second order factors, we obtain the following relations.

They are structurally similar to Equations (11a), (11b), and (11c) in Kepko et al. (1996), but different in detail:

$$B_{X'} = \frac{B_{p}(1+g^{2})}{2gG_{p}}$$

$$+ \cos\omega t \left[O_{S1} + \frac{B_{a}(\sigma_{Px} - \delta\theta_{S1})}{gG_{p}}\right]$$

$$- \sin\omega t \left[O_{S2} + \frac{gB_{a}(\sigma_{Py} - \delta\theta_{S2})}{G_{p}}\right]$$

$$5 + \cos 2\omega t \left[\frac{B_{p}(1-g^{2})}{2gG_{p}}\right]$$

$$+ \sin2\omega t \frac{B_{p}}{2G_{p}} \left[g\phi_{a} - \frac{\phi_{a}}{g} + g\delta\phi_{S12}\right]$$

$$B_{Y'} = -\frac{B_{p}}{2G_{p}} \left[\frac{1+g^{2}}{g}\phi_{a} + g\delta\phi_{S12}\right]$$

$$+ \cos\omega t \left[O_{S2} + \frac{gB_{a}(\sigma_{Py} - \delta\theta_{S2})}{G_{p}}\right]$$

$$+ \sin\omega t \left[O_{S1} + \frac{B_{a}(\sigma_{Px} - \delta\theta_{S1})}{gG_{p}}\right]$$

$$10 - \cos2\omega t \frac{B_{p}}{2G_{p}} \left[g\phi_{a} - \frac{\phi_{a}}{g} + g\delta\phi_{S12}\right]$$

$$+ \sin\omega t \left[\frac{B_{p}(1-g^{2})}{2gG_{p}}\right]$$

$$(25)$$

$$B_{Z'} = \frac{B_{a}}{G_{a}} + O_{S3}$$

$$- \cos\omega t \frac{B_{p}\sigma_{Px}}{G_{a}}$$

$$+ \sin\omega t \frac{B_{p}\sigma_{Py}}{G_{a}}$$

$$(26)$$

- 15 These equations show how the parameters affect the signal content at the spin tone harmonics in the despun measurements. The first terms in all three Equations (24), (25), and (26) are the primary measurements terms. In the spin plane, the ambient magnetic field only has a  $B_X = B_p$  component. Consequently, the first term of  $B_{X'}$  is approximately  $B_p$ , while the first term of  $B_{Y'}$  is approximately 0 as  $\phi_a \approx 0$  and  $\delta \phi_{S12} \approx 0$ . In the spin axis, we find  $B_{Z'} \approx B_a$  with  $G_a \approx 1$  and  $O_{S3} \approx 0$ . In addition, superposed first and second harmonic signals are expected as functions of the calibration parameters. The first harmonic signals
- are described by the second and third terms in Equations (24), (25), and (26). In the spin plane, Equations (24) and (25), the spin tone signals are the result of spin plane offsets  $O_{S1/2}$  and projections of the spin axis field  $B_a$  onto the spin plane. In the spin axis, Equation (26), first harmonic disturbances are due to the projection of spin plane fields onto the spin axis, the reason being an incorrect description of the spin axis direction by the angles  $\sigma_{Px/y}$ . Second harmonic signals are only expected in the despun spin plane components (fourth and fifth terms of Equations 24 and 25). These are due to a mismatch in spin plane gains
- 25 (parameter g) or an unaccounted non-orthogonality between the spin plane sensor axes S1 and S2 (parameter  $\delta \phi_{S12}$ ).

**Table 2. Parameters and favorable conditions**

10

15

| Group | Parameters                                            | Disturbances                | Conditions                         | Uncertainties                                                                                                                             |
|-------|-------------------------------------------------------|-----------------------------|------------------------------------|-------------------------------------------------------------------------------------------------------------------------------------------|
| 1     | $\sigma_{\mathrm{P}x}$ and $\sigma_{\mathrm{P}y}$     | at $\omega$ along spin axis | high $B_{\rm p}$ , low $F_{\rm a}$ | $\Delta \sigma_{{ m P}x/y} pprox F_{ m a}/B_{ m p}$                                                                                       |
| 2     | $g$ and $\delta\phi_{\rm S12}$                        | at $2\omega$ in spin plane  | high $B_{\rm p},$ low $F_{\rm 2p}$ | $\Delta g\approx F_{\rm 2p}/B_{\rm p}$ and $\Delta\phi_{\rm S12}\approx 2F_{\rm 2p}/B_{\rm p}$                                            |
| 3     | $O_{\mathrm{S1}}$ and $O_{\mathrm{S2}}$               | at $\omega$ in spin plane   | low $B_{\rm a},$ low $F_{\rm p}$   | $\Delta O_{\mathrm{S1/2}} \approx F_\mathrm{p} + B_\mathrm{a} \Delta \sigma_{\mathrm{P}x/y} + B_\mathrm{a} \Delta \theta_{\mathrm{S1/2}}$ |
| 4     | $\delta \theta_{\rm S1}$ and $\delta \theta_{\rm S2}$ | at $\omega$ in spin plane   | high $B_{\rm a},$ low $F_{\rm p}$  | $\Delta \theta_{\rm S1/2} \approx F_{\rm p}/B_{\rm a} + \Delta O_{\rm S1/2}/B_{\rm a} + \Delta \sigma_{{\rm P}x/y}$                       |

**4 Favorable Conditions for the Determination of the Calibration Parameters**

From the factors pertaining to the first and second harmonic terms of  $B_{X'}$ ,  $B_{Y'}$ , and  $B_{Z'}$  (Equations 24 to 26) it is possible to derive the conditions that should be favorable for the determination of the 8 previously mentioned parameters. These factors are:

5
$$\left[O_{\mathrm{S1}} + \frac{B_{\mathrm{a}}(\sigma_{\mathrm{P}x} - \delta\theta_{\mathrm{S1}})}{gG_{\mathrm{p}}}\right]$$
 and  $\left[O_{\mathrm{S2}} + \frac{gB_{\mathrm{a}}(\sigma_{\mathrm{P}y} - \delta\theta_{\mathrm{S2}})}{G_{\mathrm{p}}}\right]$  (27)

$$\left[\frac{B_{\rm p}\sigma_{\rm Px}}{G_{\rm a}}\right] \qquad \text{and} \qquad \left[\frac{B_{\rm p}\sigma_{\rm Py}}{G_{\rm a}}\right] \tag{28}$$

$$\left[\frac{B_{\rm p}(1-g^2)}{2gG_{\rm p}}\right] \qquad \text{and} \qquad \frac{B_{\rm p}}{2G_{\rm p}}\left[g\phi_{\rm a} - \frac{\phi_{\rm a}}{g} + g\delta\phi_{\rm S12}\right] \tag{29}$$

Here, the factors (27) and (28) pertain to the spin tone disturbances in the despun spin plane and spin axis components, respectively, and (29) pertains to the second harmonic frequency disturbance (double spin tone frequency) in the spin plane components.

As can be seen, the first factor of the latter group (29) is dependent on  $B_p$ , the external field in the spin plane which we assume to be constant, on  $G_p$ , the absolute gain factor in the spin plane which should be approximately 1, and on 1/g - g, which is 0 only if g = 1. Hence, the presence of one part of the second harmonic disturbance, though modulated by  $B_p$ , is ultimately dependent only on g, the ratio of spin plane gains. Consequently, this relation can be used to determine g correctly. The signal to do that and, in particular, the signal to noise ratio (SNR) is larger if  $B_p$  is larger. We capture this relation in the second line of Table 2. As the second harmonic disturbance in the spin plane is to be minimized to get g, the natural fluctuations around that frequency (of amplitude  $F_{2p}$ ) should also be low in the spin plane. The uncertainty in g is then expected to be on the order of  $F_{2p}/B_p$ .

The same is true for the complementary factor, on the right side of (29): Also this second harmonic disturbance is modulated

20 by  $B_p$ . When g is accurately determined, then the  $\phi_a$  influence vanishes, and the entire factor can only vanish by correctly choosing  $\delta\phi_{S12}$ . Hence, to determine this parameter accurately, also  $B_p$  should be large and the natural fluctuations at the second harmonic should be of low amplitude (low  $F_{2p}$ ). The uncertainty  $\Delta\phi_{S12}$  of  $\delta\phi_{S12}$  and, ultimately,  $\phi_{S12}$  is expected to be on the order of  $2F_{2p}/B_p$  (see line 2 in Table 2).

Let's focus on the group of factors (28). The spin frequency disturbance is clearly modulated by  $B_p$ , as  $G_a$  should be close to 1, so  $B_p$  benefits the SNR. These disturbances vanish if  $\sigma_{Px}$  and  $\sigma_{Py}$  become 0, i.e., if they are precisely determined. A low amplitude in the natural fluctuations at the spin frequency along the spin axis  $F_{\rm a}$  would also support the determination. The uncertainty in  $\sigma_{\rm Px}$  and  $\sigma_{\rm Py}$  is then expected to be on the order of  $\Delta \sigma_{\rm Px/y} \approx F_{\rm a}/B_{\rm p}$  (line 1 in Table 2).

The first set of factors in (27) pertain to the spin frequency disturbances in the spin plane components. They consist of two parts: a spin plane offset component  $O_{S1}$  or  $O_{S2}$ , and a term that is modulated by  $B_a$  and which may vanish if the difference  $(\sigma_{Px} - \delta\theta_{S1})$  or  $(\sigma_{Py} - \delta\theta_{S2})$  vanishes. Obviously, if  $B_a$  vanishes, then the spin plane spin frequency disturbances can only come from the spin plane offset components. Hence, for their determination it is beneficial if the spin axis field  $B_a$  is low and if the natural fluctuation level around the spin frequency in the spin plane  $F_p$  is low. The uncertainty in  $O_{S1}$  and  $O_{S2}$  is then expected to be on the order of  $F_p + B_a \Delta \sigma_{Px/y} + B_a \Delta \theta_{S1/2}$  (see line 3 of Table 2).

The remaining elevation angles  $\delta\theta_{S1}$  and  $\delta\theta_{S2}$  are most difficult to determine: it is beneficial if the spin axis field  $B_a$  is 10 high. In addition, however, it is necessary that the spin axis itself is well determined, as the parameters  $\sigma_{Px}$  and  $\sigma_{Py}$  equally influence the spin tone signal in the spin plane as  $\delta\theta_{S1}$  and  $\delta\theta_{S2}$ . Note that  $\sigma_{Px}$  and  $\sigma_{Py}$  can be independently determined by minimizing the spin frequency disturbances in the spin axis component.  $F_p$  should again be low. Altogether, the uncertainty in  $\delta\theta_{S1/2}$  is on the order of  $\Delta\theta_{S1/2} \approx F_p/B_a + \Delta O_{S1/2}/B_a + \Delta \sigma_{Px/y}$  (see line 4 of Table 2).

**5** Parameter Determination**

15 Based on the findings from the previous section, we propose algorithms to determine the 8 spin-related parameters in an iterative manner (sections 5.2 to 5.5). The algorithms are based on computing estimates of the parameters for short intervals, and evaluate the uncertainties of those estimates based on the uncertainties indicated in Table 2. Then, the estimates with uncertainties below a certain acceptable threshold are chosen to form the basis of one parameter correction.

**5.1 Precalibration**

20 The temperature dependent gains  $\mathbf{G}_{\mathrm{T}}(T_{\mathrm{s}}, \underline{T}_{\mathrm{e}})$  determined on ground should be used to convert the raw magnetometer output  $\boldsymbol{B}_{\mathrm{S}}$  to a precalibrated, temperature corrected intermediate product  $\boldsymbol{B}_{\mathrm{ST}}$ , according to Equation (8).

The offset vector  $O_{\rm S}$  and the calibration matrices  $\Phi$ ,  $\Sigma$ ,  $\Gamma$ , and  ${\rm G}$  should be initiated with the best known values at the time of calibration. At the beginning, these will be ground obtained values:

- for  $\theta_{S1}$ ,  $\theta_{S2}$ ,  $\phi_{S12}$ , and  $O_S$  from ground magnetometer calibration,
- 25 for  $\phi_a$  from nominal spacecraft design or mirror/laser based alignment measurements,
  - for  $\sigma_{Px}$  and  $\sigma_{Py}$  from an initial estimate of the spin axis direction (alternatively  $\sigma_{Px} = \sigma_{Py} = 0$  may be chosen),
  - and  $G_{\rm p} = G_{\rm a} = g = 1$  due to precalibration.

If in-flight calibration has already taken place, then these values will be superseded by better in-flight determined values.

**5.2 Calibration of the Spin Axis Direction**

5

15

The entire interval of magnetic field measurements should be divided into small (overlapping) subintervals of length  $t_{int} = 2\pi n/\omega$ , with  $n \in \mathbb{N}$ . The factor n should not be too small; hence, the subintervals should contain a number of spin periods, so that the spin tone at the spin frequency and also the power around that frequency can be accurately determined. On the other hand, subintervals should not be too large, so that the field/environmental conditions can be assumed constant.

For each of the subintervals, the uncertainties  $\Delta \sigma_{Px/y} \approx F_a/B_p$  need to be calculated (line 1 in Table 2). Conservatively, we choose  $B_p$  to be the minimal modulus of the spin plane field over the subinterval:  $\min\left(\sqrt{B_x^2 + B_y^2}\right)$ .  $F_a$  can be estimated by taking the maximum of the discrete Fourier components  $F_{a\pm}$  of the spin axis magnetic field  $B_z$  at frequencies  $\omega_{\pm}$  that are slightly over and under the spin frequency:  $\omega_{\pm} = 2\pi n_{\pm}/t_{int}$  with  $n_{\pm} \in \mathbb{N}$  and slightly over/under n:

10
$$F_{a\pm} = \mathcal{F}(B_z, \omega_{\pm}) = \left| \frac{2}{N} \sum_{k=0}^{N-1} B_z(t_0 + k\delta t) \exp\left(-i\omega_{\pm}k\delta t\right) \right|$$
(30)
$$F_a = \max(F_{a\pm})$$
(31)

Here  $t_0$  is the start of a subinterval considered, N is the number of magnetic field measurement samples in that subinterval, and  $\delta t$  is the sampling period. The frequencies  $\omega_+$  and  $\omega_-$  should sufficiently differ from  $\omega$  to avoid leakage from spin tone. However,  $\omega_+$  and  $\omega_-$  should also be close enough to  $\omega$  so that the amplitudes at those frequencies resemble the natural amplitude level at the spin frequency. Note that the optimal choice of  $\omega_+$  and  $\omega_-$  is subinterval specific. In practice, however,

fixed frequencies can be used that are at some distance  $|\omega_{\pm} - \omega|$ , if that distance is safely larger than usual spin tone spectral peak widths.

From here on, we use  $\mathcal{F}(B,\omega)$  to denote the Fourier component of B at frequency  $\omega$ . It should be noted that it may be recommended to de-trend the B data before computing  $\mathcal{F}(B,\omega)$ , by simply subtracting a linear fit. Linear trends will not occur

20 if the external field can be assumed to be constant. In many real applications, however, the spacecraft will move through field gradients during subintervals considered, and in these cases, the linear trend in the field measurements will increase the spectral content across the spectrum.

Parameter estimates  $\sigma_{Px}$  and  $\sigma_{Py}$  are determined by minimization of the spin tone  $S_a$  in the spin axis component:  $S_a = \mathcal{F}(B_z, \omega)$ . This minimization is performed for each subinterval. Hence, we obtain for each subinterval one estimate for  $\sigma_{Px}$ , 25 for  $\sigma_{Py}$ , and for the uncertainty  $\Delta \sigma_{Px/y}$ . A final parameter update for  $\sigma_{Px}$  and  $\sigma_{Py}$  
[revised manuscript text omitted]